# SPA: A Simple but Tough-to-Beat Baseline for Knowledge Injection

Kexian Tang [* 1]   Jiani Wang [* 1]   Shaowen Wang [1]   Kaifeng Lyu [† 1]

## Abstract

While large language models (LLMs) are pre-trained on massive amounts of data, their knowledge coverage remains incomplete in specialized, data-scarce domains, motivating extensive efforts to study synthetic data generation for knowledge injection. We propose **SPA** (**S**caling **P**rompt-engineered **A**ugmentation), a simple but tough-to-beat baseline that uses a small set of carefully designed prompts to generate large-scale synthetic data for knowledge injection. Through systematic comparisons, we find that **SPA** outperforms several strong baselines. Furthermore, we identify two key limitations of prior approaches: (1) while RL-based methods may improve the token efficiency of LLM-based data augmentation at small scale, they suffer from diversity collapse as data scales, leading to diminishing returns; and (2) while multi-stage prompting may outperform simple augmentation methods, their advantages can disappear after careful prompt tuning. Our results suggest that, for knowledge injection, careful prompt design combined with straightforward large-scale augmentation can be surprisingly effective, and we hope SPA can serve as a strong baseline for future studies in this area. Our code is available at https://github.com/Tangkexian/SPA.

## 1. Introduction

Large language models (LLMs) can acquire broad world knowledge and general capabilities from massive web text, but their coverage of specialized domains is often incomplete. A common approach to address this issue is to inject domain knowledge into these models by further fine-tuning or continually pretraining them on domain-specific data that are not readily available on the public web (Singhal et al., 2023; Wu et al., 2023; Zhang et al., 2024).

*Equal contribution [1]Institute for Interdisciplinary Information Sciences, Tsinghua University, Beijing, China. Correspondence to: Kaifeng Lyu <klyu@mail.tsinghua.edu.cn>.

*Proceedings of the $43^{rd}$ International Conference on Machine Learning*, Seoul, South Korea. PMLR 306, 2026. Copyright 2026 by the author(s).

However, domain-specific datasets are usually limited in both scale and diversity. In contrast to the high redundancy of web text, domain knowledge often lacks the repetitive variations required for effective learning. Compounding this challenge, LLMs are known to be data-inefficient, necessitating frequent and varied exposure to facts for reliable internalization (Kandpal et al., 2023; Allen-Zhu & Li, 2025; Abonizio et al., 2025). Consequently, directly fine-tuning LLMs on such sparse data often leads to overfitting on specific surface forms rather than robust knowledge acquisition, causing the model to fail when encountering unseen queries or variations (Berglund et al., 2024; Shumailov et al., 2023).

To mitigate this scarcity, an increasingly popular paradigm is to expand the domain-specific data into a much larger synthetic corpus using an LLM-based data generator, and then train the target model on this augmented dataset. Recent efforts to design such augmentation methods generally follow two distinct methodologies. RL-based approaches, such as SEAL (Zweiger et al., 2025), employ reinforcement learning (RL) to train a dedicated generator, where the reward is defined as the performance of the target model trained on the augmented data. Another line of works, such as Enti-Graph (Yang et al., 2025) and Active Reading (Oguz et al., 2026), relies on prompting pipelines consisting of multiple stages, where the data generator is prompted to rewrite the original corpus into a high-quality synthetic corpus through a sequence of intermediate transformations.

In this work, we aim to push the frontier of synthetic data generation for knowledge injection by proposing a simple but tough-to-beat baseline method: **SPA** (**S**caling **P**rompt-engineered **A**ugmentation), which is a knowledge injection method that consistently outperforms existing methods at scale (Figure 1). The key idea of SPA is very simple:

1. **Prompt Engineering**: We draw upon insights from cognitive science and educational psychology to design a set of 7 prompt templates based on effective human learning strategies, including *concept learning*, *critical thinking*, and *generative learning*.

2. **Scaling**: We repeatedly prompt an LLM to rewrite the source content using templates from our prompt set, progressively *scaling* the augmented corpus into a large-scale synthetic corpus.

3. **Training**: Finally, we train the target model on the synthetic corpus following the same settings as prior work.

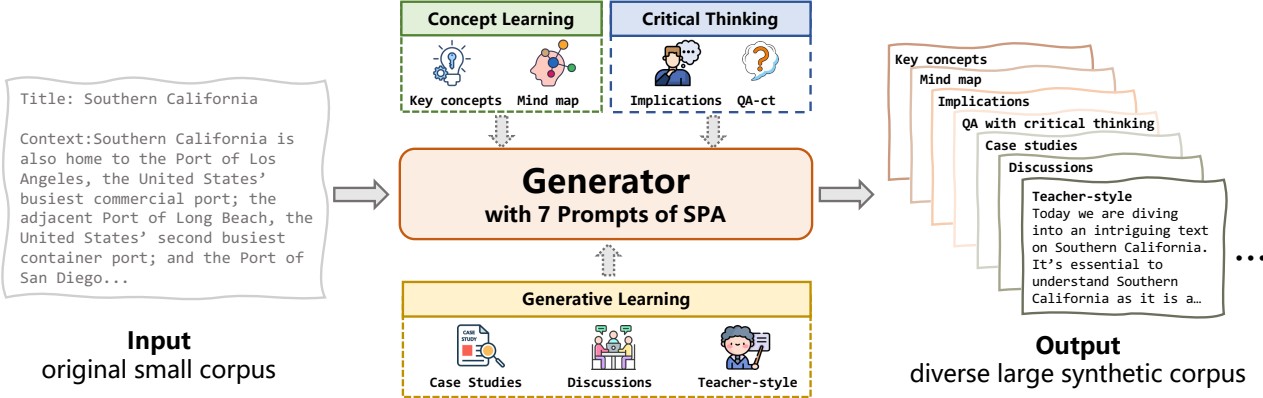

*Figure 1.* **Overview of Scaling Prompt-engineered Augmentation (SPA).** Our baseline method rewrites a small source corpus into a large synthetic corpus by repeatedly prompting a generator with a fixed set of seven human-curated prompt templates, which are designed based on three levels of learning strategies: Concept Learning, Critical Thinking, and Generative Learning.

Perhaps surprisingly, we find that this simple baseline is already highly effective. We evaluate SPA on three representative benchmarks that span Wikipedia-based question answering (SQuAD, Rajpurkar et al. 2016), long-document comprehension (QuALITY, Pang et al. 2022), and multi-hop reasoning (MultiHop-RAG, Tang & Yang 2024). Across all these settings, we use almost the same experimental setup as in the previous works (Zweiger et al., 2025; Yang et al., 2025), and change only the data augmentation strategy. We observe that SPA consistently matches or outperforms a range of substantially more complex methods, including SEAL (Zweiger et al., 2025), PaST (Tang et al., 2026), Enti-Graph (Yang et al., 2025), SoG (Ma et al., 2025), and Active Reading (Oguz et al., 2026), and the gains of SPA become more pronounced as we scale the synthetic corpus.

**Our contributions are:**

- We propose **S**caling **P**rompt-engineered **A**ugmentation (SPA), a simple knowledge injection baseline that uses a small set of prompt templates to generate large-scale synthetic corpora (Section 3).

- We conduct a series of strictly controlled, token-matched evaluations comparing SPA against many complex methods (Section 4), and demonstrate that SPA consistently matches or outperforms these methods at scale (Section 5). This suggests that SPA is a competitive baseline for knowledge injection despite its simplicity.

- Our results provide insights into the limitations of existing methods: (1) RL-based methods suffer from diversity collapse as data scales, leading to diminishing returns; and (2) adding more stages in the prompting pipeline may not yield larger gains than improving the quality of the prompt set for one-stage methods.

While SPA demonstrates strong effectiveness, we do not suggest that knowledge injection should just rely on SPA. Rather, we hope our results provide a solid foundation for future work to develop methods that genuinely improve upon competitive baselines.

## 2. Related Work

**Small Corpus Knowledge Injection.** Knowledge injection for domain-specific data has a wide range of applications. For example, in domains such as medicine (Peng et al., 2019; Lee et al., 2019; Luo et al., 2022), finance (Araci, 2019; Liu et al.; Li et al., 2023), and law (Chalkidis et al., 2020; Guha et al., 2023), it is often necessary to inject specialized knowledge into large language models to support professional tasks. However, domain-specific corpora typically suffer from data scarcity. As a result, directly performing continued pretraining or fine-tuning on such corpora often exhibits limited effectiveness and poor scalability (Gu et al., 2025; Kandpal et al., 2023; Allen-Zhu & Li, 2025; Petroni et al., 2019; Sun et al., 2024). To address this issue, our method SPA aims to scale up small corpora by synthetic data generation based on the original data, following recent advances in this line of work (Zweiger et al., 2025; Ovadia et al., 2024; Tang et al., 2026; Mecklenburg et al., 2024; Yang et al., 2025; Oguz et al., 2026; Ma et al., 2025).

**Synthetic Data Generation.** Prior work has explored various strategies for synthetic generation of pretraining data, including (1) prompting LLMs to rewrite documents in diverse styles such as children's stories (Eldan & Li, 2023), "Wikipedia-style" data, and "question–answer" formats (Maini et al., 2024), and (2) directly prompting LLMs to generate synthetic corpora such as textbook-style data (Gunasekar et al., 2023). However, different from synthesizing data from large pretraining corpora, generating data from small corpora is more challenging. Such corpora typically lack the scale and redundancy needed for effective learning, requiring higher diversity and broader coverage in the generated data. A more detailed discussion of this line of research is provided in Section 3.2.

**Knowledge Editing.** A related line of work focuses on localizing factual knowledge within LLMs and then updating the model (Meng et al., 2022; Mitchell et al., 2022;

Meng et al., 2023) or maintaining an external module to edit specific facts with minimal side effects (Zhong et al., 2023; Cohen et al., 2024). This line of research, known as knowledge editing, differs from our setting. In contrast, we consider *corpus-level* knowledge injection, where the model is required to learn from a corpus of documents and utilize many interrelated facts expressed in diverse forms, rather than isolated fact tuples.

**Retrieval-Augmented Generation.** Retrieval-augmented generation (RAG) is a widely adopted alternative to parametric updating, as it enables models to retrieve relevant information from external knowledge bases before answering questions (Lewis et al., 2020; Borgeaud et al., 2022; Izacard et al., 2023; Gao et al., 2023; Asai et al., 2024). In our setting, we do not aim to position our approach as a replacement for RAG, but rather as a complementary paradigm to knowledge accessed through retrieval-augmented generation (Yang et al., 2025).

## 3. SPA: Scaling Prompt-engineered Augmentation

We focus on the problem of injecting a small domain-specific text corpus into a pretrained language model. Our goal is to enable the model to internalize this knowledge in its parameters, so that it can correctly answer domain-relevant questions without access to the original corpus at inference time. Following EntiGraph (Yang et al., 2025), we consider synthetic data generation as an effective method for knowledge injection in data-scarce regimes: the original niche corpus is first expanded into a much larger synthetic training set, which is then used for continued pretraining.

### 3.1. Problem Setup

Let $\mathcal{D} = \{d_i\}_{i=1}^N$ denote a small domain-specific corpus, where each $d_i$ corresponds to a single document. Our objective is to design a data augmentation algorithm $\mathcal{A}$ that transforms $\mathcal{D}$ into a larger synthetic corpus $\tilde{\mathcal{D}} = \mathcal{A}(\mathcal{D})$, which is subsequently used for more effective knowledge injection through continued pretraining. In this work, we focus on *task-agnostic* augmentation strategies that operate directly on the original documents and do not require access to downstream questions or labels. This design makes our approach broadly applicable and transferable to other knowledge-injection scenarios.

### 3.2. Existing Methods

Recent work on synthetic data generation for knowledge injection can be divided into two categories: *RL-based augmentation methods* and *multi-stage prompting methods*.

**RL-based augmentation methods.** This line of research employs reinforcement learning to train a dedicated data generator that produces synthetic data conditioned on the original corpus.

*SEAL* (Zweiger et al., 2025) consists of three main stages: (i) *Data generation and training*: the data generator generates synthetic data conditioned on the source passages, then the target model is fine-tuned on this data; (ii) *Reward signal estimation*: the downstream QA accuracy of the target model is used as a reward signal to indicate which types of data are more effective; (iii) *Iterative improvement*: the data generator is updated based on the reward signal, and the process is repeated iteratively to improve the quality of the synthetic data. In SEAL's implementation, the data generator shares the same base model as the target model. This iterative procedure encourages the generator to produce data that is easier for the target model to absorb, enabling SEAL to achieve strong performance in low-data settings, even outperforming data generated by more powerful models such as GPT-4.1. While RL can be effective, the computational burden of training makes it impractical to apply RL to every new domain. To mitigate this issue, *PaST* (Tang et al., 2026) (i) extracts a skill vector for knowledge manipulation from an RL-trained model in a small scale corpus, and (ii) linearly injects this vector into the generation model to enhance its ability in knowledge manipulation.

However, these methods typically rely on downstream-task performance as the RL signal. Such supervision is often unavailable in realistic settings, where no specific downstream task may be defined at the time of knowledge injection. Our method SPA focuses on *task-agnostic* augmentation strategies which do not require access to downstream tasks.

**Multi-stage prompting methods.** This line of research introduces multi-stage augmentation pipelines, where the transformation from the original corpus to the final synthetic data proceeds through several intermediate steps.

*EntiGraph* (Yang et al., 2025) first breaks down the original corpus into a list of entities, and then uses a language model to generate text descriptions about relations among the extracted entities. By rearranging the knowledge in this way, EntiGraph expands a data-scarce corpus and injects it into an 8B model, leading to strong downstream performance that exceeds GPT-4's performance. *SoG* (Ma et al., 2025) and *GraphGen* (Chen et al., 2025) also follow graph-based designs, constructing a graph over corpus chunks and leveraging this structure to guide data generation. *Active Reading* (Oguz et al., 2026) also follows a multi-stage design: it first prompts the generator to produce document-specific learning strategies, such as paraphrasing, knowledge linking, and analogical reasoning; then it uses these strategies together with the original corpus to generate augmented data. This approach yields improved accuracy for 8B-scale models on a Wikipedia-grounded subset of SimpleQA, even exceeding the performance of models that are provided with the full context at test time.

However, such pipelines can increase system complexity and may require careful optimization at every stage. For example, as analyzed in Section 5.2, the first-stage strategy generation in Active Reading appears less effective than SPA, which could lead to its weaker overall performance.

### 3.3. Our Method: SPA

In this paper, we propose the method of **S**caling **P**rompt-engineered **A**ugmentation (SPA) to serve as a simple but tough-to-beat baseline for knowledge injection.

**Our Method.** Our method SPA rewrites the original corpus into a large synthetic corpus by repeatedly prompting a language model $G$ with a fixed set of human-curated prompt templates $\{P_i\}_{i=1}^M$, where $M = 7$. Specifically, given a target number of tokens $\tilde{\mathcal{D}}$, we generate data independently with each prompt template $P_i$. Each prompt $P_i$ is used to generate the same amount of tokens, which is approximately of size $\tilde{\mathcal{D}}/M$. Then the synthetic data generated by all $M$ prompts are combined into the final synthetic corpus of size $\tilde{\mathcal{D}}$. We argue that applying this baseline method at scale (i.e., for large $\tilde{\mathcal{D}}$) produces synthetic data that is effective for knowledge injection and can outperform existing methods mentioned in Section 3.2.

**Prompt Engineering.** A key component of our baseline method is the design of the prompt set $\{P_i\}_{i=1}^M$. For this, we draw upon insights from cognitive science and educational psychology to identify three levels of learning strategies that are effective for human learning: **Concept Learning**, **Critical Thinking**, and **Generative Learning**.

**Level 1: Concept Learning.** Concept learning is a strategy that requires learners to search for and test attributes that can be used to distinguish exemplars of a concept from non-exemplars (Bruner et al., 1956). Motivated by its effectiveness for human learning, and in particular by prior work showing that concept mapping and mind maps help learners organize and externalize conceptual structures (Novak & Gowin, 1984), we design two prompt templates for concept learning: (1) *Key concepts*: A prompt template that asks LLMs to identify and explain the key concepts in the given text; (2) *Mind map*: A prompt template that asks LLMs to create a mind map that organizes the key concepts.

**Level 2: Critical Thinking.** Critical thinking refers to the process of systematically analyzing facts, evidence, observations, and arguments to arrive at well-reasoned conclusions, which has been shown to encourage deep comprehension (Brookfield, 1987; Bransford et al., 2000). We design two prompt templates: (3) *Implications*: A prompt template that asks LLMs to infer direct and indirect consequences from the given text to extend beyond explicit statements, and (4) *QA-ct (QA with critical thinking)*: A prompt template that asks LLMs to generate in-depth question-answer pairs that require high-order critical thinking.

**Level 3: Generative Learning.** Generative learning is a strategy that requires learners to actively make sense of learning material so that they can apply it to new situations (Wittrock, 1974; Fiorella & Mayer, 2015). In this spirit, we design three prompt templates for generative learning: (5) *Case studies*: A prompt template that asks LLMs to generate a case study for the given text; (6) *Discussions*: A prompt template that asks LLMs to generate a natural in-depth discussion dialogue between two readers about the content of the provided text; and (7) *Teacher-style*: A prompt template that asks LLMs to explain the text like a knowledgeable teacher.

To mitigate hallucination, we further add a constraint in the templates to strictly adhere to the provided text. See Appendix A and D for full prompts and generation examples.

**Comparison to Prior Work.** Compared to prior work, EntiGraph can be viewed as a multi-stage prompting approach to encourage concept learning, as it requires the generator to discuss the relationships between entities in the text. SEAL uses *Implications* as their base prompt template and use RL to improve data generation quality. Active Reading relies on LLMs themselves to come up with document-specific learning strategies for the document to be augmented. In contrast, SPA is a simple single-stage prompting approach that is based on high-quality human-curated prompt templates by drawing upon insights from cognitive science and educational psychology, covering three levels of learning strategies.

## 4. Experimental Setup

We study three representative knowledge injection settings that align with prior work:

**(A) SQuAD: Wikipedia-based Question Answering.** SQuAD (Rajpurkar et al., 2016) is a reading comprehension dataset consisting of over 100,000 questions posed by crowdworkers on a set of Wikipedia articles, where the answer to each question is a segment of text from the corresponding reading passage. Following the setting of SEAL, we use a subset of $n = 200$ passages from SQuAD as the original corpus to generate synthetic data for continued pre-training. Both the generator and the model to be adapted are Qwen2.5-7B, following SEAL's setup (Section 3.2). The adapted model is then evaluated on 974 questions related to the original passages without providing the passages at test time. Additional training and evaluation details are provided in Appendix B.1 and C.1.

One key difference from SEAL is that we substantially scale up the amount of synthetic data. In SEAL, only 5 augmented samples are generated per passage, which we find insufficient for effective knowledge injection: with this setting the model reaches only 58.2% accuracy, whereas increasing the number of samples to 27 improves performance to

70.74%. In our experiments, to further explore the upper bound of performance, we progressively scale the amount of synthetic data up to 3200 augmented samples per passage. This corresponds to approximately 120M tokens in total and $4000\times$ the token count of the original corpus.

For baselines in this setting, we include (1) SEAL (Zweiger et al., 2025) and (2) PaST (Tang et al., 2026). Since PaST does not release its code or trained models, we report the best performance numbers from their paper. We also include (3) Active Reading (Oguz et al., 2026), which reports strong results for knowledge injection on Wikipedia-based corpora. Active Reading considers two variants: a task-agnostic setting and a task-specific setting that assumes access to the downstream task. To ensure a fair comparison, we adopt the task-agnostic variant throughout this paper. In addition to these methods, we include two commonly used simple but strong baselines, (4) Rephrase and (5) QA (Maini et al., 2024). For all methods, we match the total number of training tokens to ensure fair comparisons.

**(B) QuALITY: Long-Document Comprehension.** QuALITY (Pang et al., 2022) is a multiple-choice question-answering dataset with context passages in English that have an average length of about 5,000 tokens. Following EntiGraph, we use 265 passages from QuALITY as the original corpus. We adapt Meta-Llama-3-8B (Grattafiori et al., 2024) and use gpt-oss-120b (Agarwal et al., 2025) as the generator model to produce synthetic data. We scale the synthetic training data to 455M tokens to match EntiGraph's setting, which corresponds to roughly $350\times$ the token count of the original corpus. We evaluate on the questions associated with the original passages without providing the passages at test time. More training and evaluation details are provided in Appendix B.2 and Appendix C.2.

For baselines in this setting, we include (1) EntiGraph (Yang et al., 2025), (2) Active Reading (Oguz et al., 2026), and (3) SoG (Ma et al., 2025). Since SoG does not release its code or trained models, we report the best performance numbers from their paper. In addition to these methods, we include a simple (4) QA baseline, which EntiGraph identifies as a strong method in its appendix. We do not include Rephrase, as EntiGraph reports that it underperforms relative to EntiGraph on QuALITY.

**(C) MultiHop-RAG: Multi-Hop Queries.** To provide a broader and more general evaluation setting, we consider multi-hop reasoning, which differs substantially from the two previous tasks. We use MultiHop-RAG (Tang & Yang, 2024) as our evaluation dataset. MultiHop-RAG consists of a collection of multihop queries, ground-truth answers, the associated supporting evidence and corresponding full passage. We use the entire MultiHop-RAG corpus as the original corpus, which contains 609 articles. We scale the resulting synthetic data to 15M tokens. We do not further

increase the scale, as at this level our method already exhibits a clear performance gap over the baselines. We use GPT-4o-mini (Hurst et al., 2024) as the generator model and adapt both Meta-Llama-3-8B and Qwen2.5-7B. We evaluate the adapted models on the open-ended questions of MultiHop-RAG without providing the original passages, using zero-shot prompting. More training and evaluation details are provided in Appendix B.3 and C.3. For baselines in this setting, we include two strong baselines: (1) EntiGraph and (2) Active Reading.

*Table 1.* **Main results at the largest synthetic token scale on each benchmark.** QA accuracy (%) is reported across SQuAD, QuALITY, and MultiHop-RAG, with shared settings grouped. For each benchmark, *Model* refers to the adapted model, *Generator* refers to the model used for synthetic data generation, and *Number of Tokens* refers to the total synthetic token budget. Each value corresponds to the final performance at the largest token scale used in that benchmark. ∗ Base denotes the model without additional training, ⋆ denotes simple baselines, ▷ refers to prior high-performing methods, and ▶ denotes our method SPA.

| SQuAD | QA Accuracy(%) |
| --- | --- |
| *Model: Qwen2.5-7B* | |
| *Generator: Qwen2.5-7B (self-generated)* | |
| *Number of Tokens: 120M* | |
| ∗ Base | 31.31 |
| ⋆ Rephrase | 86.86 |
| ⋆ QA | 89.63 |
| ▷SEAL | 74.23 |
| ▷Active Reading | 90.25 |
| ▶SPA | **91.27** |

| QuALITY | QA Accuracy |
| --- | --- |
| *Model: Meta-Llama-3-8B* | |
| *Generator: gpt-oss-120b* | |
| *Number of Tokens: 455M* | |
| ∗ Base | 39.27 |
| ⋆ QA | 52.33 |
| ▷ EntiGraph | 56.22 |
| ▷ Active Reading | 51.75 |
| ▶ SPA | **57.03** |

| MultiHop-RAG | QA Accuracy |
| --- | --- |
| *Generator: GPT-4o-mini* | |
| *Number of Tokens: 15M* | |
| *Model: Qwen2.5-7B* | |
| ∗ Base | 60.91 |
| ▷ EntiGraph | 85.42 |
| ▷ Active Reading | 83.33 |
| ▶ SPA | **86.64** |
| *Model: Meta-Llama-3-8B* | |
| ∗ Base | 73.16 |
| ▷ EntiGraph | 84.31 |
| ▷ Active Reading | 84.44 |
| ▶ SPA | **88.36** |

## 5. Main Results

**SPA consistently achieves the strongest performance across all three benchmarks.** Table 1 reports results

on SQuAD, QuALITY, and MultiHop-RAG at the largest synthetic data scale in each setting. Across all three datasets, SPA outperforms by clear margins. On SQuAD, SPA reaches 91.27%, exceeding Active Reading (90.25%) and SEAL (74.23%). On QuALITY, SPA attains 57.03%, improving over EntiGraph (56.22%) and Active Reading (51.75%). On MultiHop-RAG, SPA achieves 86.64% on Qwen2.5-7B and 88.36% on Meta-Llama-3-8B, outperforming all baselines in both cases. We will analyze the scaling behavior in more detail in Section 5.1 and Section 5.2.

**SPA is robust across different generation models and adapted model families.** Across the three benchmark settings, we employ different generation models and adapt base models from different families, yet SPA consistently achieves the strongest performance. These results indicate that SPA is not tied to particular configurations, but generalizes well across tasks, generators, and model families.

## 5.1. SQuAD

**SPA exhibits strong and consistent scaling behavior on SQuAD.** As shown in Figure 2, although SPA does not lead in the low-token regime, it *consistently improves with scale and achieves the highest accuracy at moderate-to-large token budgets.* To explore the impact at larger data scales, we increase the synthetic corpus up to 120M tokens. This scale already approaches a natural knowledge-injection target: when evaluated with access to the original passages, Qwen2.5-7B achieves 91.38% accuracy, while SPA reaches 91.27% accuracy at 120M tokens even when evaluated without access to the passages.

Compared with SEAL, although SPA slightly underperforms SEAL at small scales, it surpasses SEAL once the synthetic corpus becomes slightly larger (e.g., with 27 augmented samples per passage). As the scale increases further, SEAL's performance saturates while SPA continues to improve, ultimately outperforming SEAL by a large margin at 120M tokens (91.27% vs. 74.23%).

Compared with Active Reading, SPA matches or even outperforms its performance across most scaling regimes. Notably, Active Reading continuously generates document-specific strategies as the synthetic corpus grows, whereas SPA relies on a fixed set of seven prompts. Moreover, SPA adopts a single-stage augmentation pipeline, making it both simpler and more effective than Active Reading, which is a multi-stage method.

**RL-based augmentation saturates at larger scales, while SPA continues to improve.** At smaller synthetic budgets, SEAL initially outperforms both SPA and Active Reading, which is expected since it is trained with reinforcement learning using downstream-task performance as the reward signal. However, as the amount of synthetic data increases, SEAL's gains begin to saturate, while SPA and Active Read-

ing continue to benefit from scaling. With even a modest increase in scale, both methods overtake SEAL and achieve better overall performance. Notably, both SPA and Active Reading are task-agnostic and have no access to downstream supervision during data generation. We provide more analysis in Section 6.1.

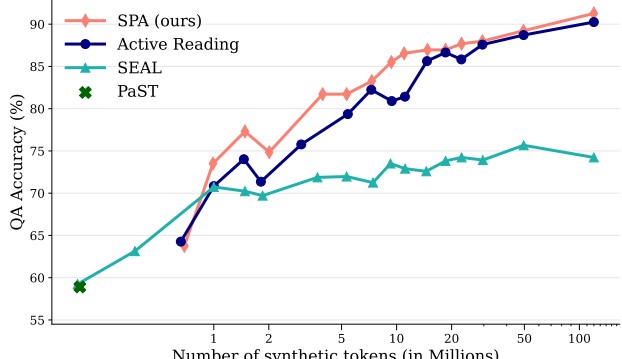

*Figure 2.* **Scaling Curve on SQuAD shows that SPA exhibits strong and consistent scaling behavior.** The y-axis represents QA accuracy, and the x-axis represents the synthetic token budget. *Note*: the PaST data point corresponds to the best performance reported in the original paper.

## 5.2. QuALITY

**A weaker but cheaper generator is sufficient for SPA to outperform prior methods.** EntiGraph uses GPT-4-Turbo to generate all synthetic data. Although GPT-4-Turbo is a strong generator, its API cost makes it impractical for large-scale data generation. We therefore ask *whether SPA can remain competitive when using a smaller and more economical generator*. To this end, we adopt gpt-oss-120b, which activates only 5.1B parameters and is substantially more efficient at inference time. The API cost of gpt-oss-120b is approximately $50\times$ lower than that of GPT-4-Turbo.

*Table 2.* **QA accuracy of SPA and QA baseline using different generation models.** The training token here is 27M.

| Method | Generation Model | QA Accuracy(%) |
|--------|------------------|----------------|
| QA | GPT-4-Turbo | 52.99 |
| QA | gpt-oss-120b | 47.47 |
| SPA | gpt-oss-120b | **52.26** |
| QA | GPT-4o | 51.31 |
| SPA | GPT-4o | **55.49** |

To verify that gpt-oss-120b is not a stronger generator in this setting, we conduct controlled experiments at 27M token scale. We compare against the QA baseline, which EntiGraph reports as stronger than EntiGraph itself at this budget. As shown in Table 2, replacing GPT-4-Turbo with gpt-oss-120b for QA generation leads to a clear drop in accuracy (52.99% → 47.47%), indicating that *gpt-oss-120b is weaker than GPT-4-Turbo for this task*.

When both methods use gpt-oss-120b as the generator, SPA achieves 52.26% accuracy, outperforming the QA baseline (47.47%) by a substantial margin and closely approaching

the performance of the QA baseline generated with the stronger GPT-4-Turbo (52.99%).

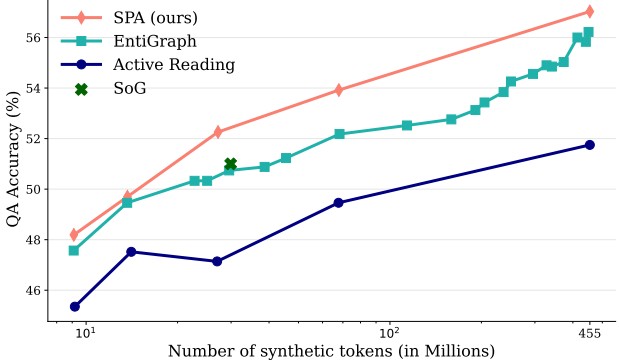

*Figure 3.* **Scaling Curve on QuALITY shows that SPA achieves the strongest scaling performance among compared methods as data scales.** The y-axis is QA accuracy, and the x-axis is the token budget. *Note*: For EntiGraph, we use statistics from the original paper, where GPT-4-Turbo is used as the generator, whereas SPA and Active Reading use gpt-oss-120b. The SoG data point corresponds to the best performance reported in the original paper, which uses a stronger base model Llama-3.1-8B-Instruct.

**SPA remains effective when using different generators.** We further replace gpt-oss-120b with GPT-4o to examine the robustness of SPA with respect to the choice of generator. Under this setting, SPA again outperforms QA baseline (55.49% vs. 51.31%), indicating that SPA benefits from stronger generators while preserving its advantage over competing methods.

**SPA continues to improve with scale and achieves the best final performance on QuALITY.** Figure 3 shows that SPA improves steadily with scale and ultimately achieves the highest QA accuracy among all methods. Notably, although gpt-oss-120b is a weaker generator than GPT-4-Turbo for this task as shown in the previous experiment, SPA at large scale (57.03%) surpasses EntiGraph generated with GPT-4-Turbo (56.22%). Moreover, when using the same generator, SPA substantially outperforms Active Reading (57.03% vs. 51.75%).

**Active Reading underperforms SPA due to lower average strategy effectiveness.** To further analyze why Active Reading underperforms SPA on QuALITY, we conduct a document-level comparison of the effectiveness of individual augmentation strategies. We randomly select five documents from the QuALITY corpus. For each document, Active Reading is first used to generate a document-specific strategy, which is then applied in the second stage to produce synthetic data. A separate model is trained on the data generated by each individual strategy and evaluated on the corresponding document's QA set without access to the original passage.

For comparison, we perform the same experiment using SPA by generating synthetic data with each of its 7 prompts inde-

pendently and training models under identical settings. The results are summarized and visualized in Figure 4. Across all 5 documents, SPA's prompts consistently achieve higher average QA accuracy than the strategies produced by Active Reading. While some Active Reading strategies perform competitively, many fall below the base model or yield only marginal improvements, as shown in Figure 4.

These results suggest that Active Reading suffers from lower average strategy effectiveness. When weaker strategies are used in the second stage to guide data generation, they may have limited impact or even introduce negative effects, which degrades the overall quality of the synthetic corpus. In contrast, SPA's human-curated prompts provide more stable and consistently effective supervision. Moreover, its single-stage design avoids the risk of low-quality intermediate outputs, leading to more reliable knowledge injection.

### 5.3. MultiHop-RAG

Table 1 reports results on MultiHop-RAG for two base models from different model families.

**SPA consistently achieves the best performance across model families.** For Qwen2.5-7B, SPA improves over the base model by 25.73 points (60.91% → 86.64%) and outperforms both Active Reading (83.33%) and EntiGraph (85.42%). For Meta-Llama-3-8B, SPA reaches 88.36%, exceeding Active Reading (84.44%) and EntiGraph (84.31%). These improvements across two different model families indicate that SPA does not rely on properties of a particular backbone, but generalizes across architectures.

**SPA generalizes to multi-hop reasoning tasks.** Compared to single-document settings, MultiHop-RAG requires models to integrate and reason over information from multiple passages. The strong improvements achieved by SPA in this setting demonstrate that its prompt-based augmentation strategy can effectively transfer to multi-hop knowledge injection scenarios.

## 6. Analysis

### 6.1. Synthetic Data Diversity

**Motivation.** To better understand why the RL-based method SEAL exhibits performance saturation as the token budget increases on SQuAD, we first conducted a qualitative inspection of its generated samples. We observed that many outputs were highly similar in both content and format, suggesting limited diversity. Since diversity is correlated with downstream performance (Havrilla et al., 2024), we then performed a systematic comparison of diversity across methods.

**Evaluation Setup.** We use the diversity evaluation toolkit of Shaib et al. (2024a), which quantifies both lexical and syntactic diversity in generated text. We report four diversity metrics following Shaib et al. (2024a). For lexical diversity,

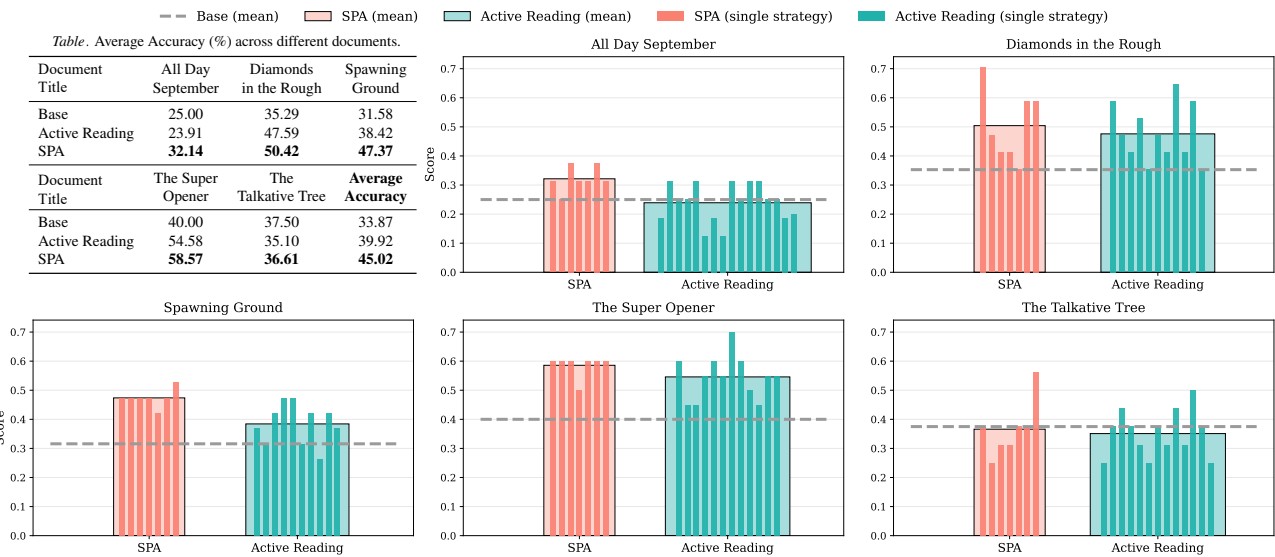

*Figure 4.* **Document-level comparison shows that SPA achieves higher average strategy effectiveness than Active Reading on QuALITY.** The table reports the average accuracy (%) of each method across all strategies for each document, showing that SPA consistently attains higher mean accuracy than Active Reading. Bold numbers indicate cases where SPA outperforms Active Reading. The subplots visualize the accuracy of individual strategies for each document, including seven strategies for SPA and a variable number for Active Reading. In each subplot, wide bars denote the average accuracy across all strategies within each method, and the narrow bars denote the accuracy of individual strategies. The gray dashed line denotes the base model's accuracy.

we use *Compression Ratio*, *Self-Repetition*, and *Self-BLEU*, which have low mutual correlation. To measure syntactic diversity, we report *Compression Ratio over part-of-speech sequences (CR-POS)*. Prior work shows that CR-POS effectively distinguishes highly diverse human-written text from lower-diversity model-generated text (Shaib et al., 2024a), and captures repetition of syntactic templates independent of lexical content (Shaib et al., 2024b). Definitions of the four metrics are provided in Appendix C.4.

We conduct the evaluation using five source articles per benchmark. Here we report results on SQuAD in the main paper. Full results for all benchmarks and additional implementation details are provided in Appendix C.4.

**RL-based augmentation exhibits diversity collapse.** As shown in Table 3, the RL-based method SEAL exhibits substantially lower diversity than all other methods across all metrics on SQuAD. This suggests that iteratively training the data generator on selected augmented data can induce a form of ***diversity collapse***, in which the generated synthetic data converges to a narrow set of repetitive patterns. This behavior provides an explanation for SEAL's rapidly saturating scaling curve on downstream tasks (Figure 2).

**SPA achieves competitive diversity.** SPA substantially outperforms single-prompt baselines (Rephrase and QA) and the RL-based method SEAL across all four diversity metrics. Compared with Active Reading, which continuously generates different document-specific augmentation strategies as the synthetic corpus scales, SPA relies on a fixed set of seven human-curated prompts shared across all

documents. Despite this simpler design, SPA achieves diversity scores comparable to Active Reading (Table 3) and attains the best downstream performance (Figure 2).

*Table 3.* **Diversity evaluation of synthetic data** generated by different data augmentation methods on the SQuAD benchmark. Lower values indicate higher diversity. All scores are averaged across five documents sampled with a fixed random seed. Best results are in **bold**, and second-best are underlined.

| Method | CR ↓ | Self-Rep. ↓ | Self-BLEU ↓ | CR: POS ↓ |
|---|---|---|---|---|
| SEAL | 19.2468 | 8.2446 | 0.0058 | 21.2072 |
| Rephrase | 5.1330 | 5.6019 | 0.0014 | 7.0200 |
| QA | 8.9630 | 6.9882 | 0.0030 | 9.7534 |
| Active Reading | **4.1788** | **4.5854** | **0.0010** | **5.9388** |
| SPA (ours) | 4.3806 | 4.8207 | **0.0010** | 6.0024 |

### 6.2. Effects of Individual Prompts in SPA

**Individual prompts are insufficient to match the full SPA configuration.** To better understand the contribution of each prompt in SPA, we conduct an ablation study by generating synthetic data using only one prompt at a time. For each prompt, we train a separate model using the corresponding synthetic data and evaluate its performance on SQuAD. We keep the total training budget fixed at 22M tokens for each prompt. The results are reported in Table 4.

The results show that models trained with individual prompts consistently underperform the full SPA configuration, with relative performance drops ranging from 2.16% to 8.73%. This result indicates that the gains of SPA do not arise from any single prompt in isolation, but rather from the combination of diverse prompts that provide complementary forms of knowledge transformation.

**Even the weakest standalone prompt remains important in the full prompt set.** Among all prompts, *Key concepts* exhibits the weakest standalone performance, with a drop of 8.73% when used alone. To further examine its role within the full prompt set, we conduct prompt ablation experiments on SQuAD by removing the *Key concepts* prompt from the full set while keeping the total data budget fixed at 22M tokens. As shown in Table 5, removing the *Key concepts* from the full prompt set leads to a 1.62% drop in QA accuracy relative to using all seven prompts, indicating that even the weakest standalone prompt still contributes to the overall performance of SPA when combined with other prompts.

*Table 4.* **Performance of individual SPA components trained in isolation.** All models are trained with approximately 22M tokens on SQuAD. Numbers in parentheses indicate the relative performance drop compared to full SPA.

| Method | QA Acc. (%) | Method | QA Acc. (%) |
|---|---|---|---|
| Key concepts | 78.95 (-8.73) | Case studies | 83.57 (-4.11) |
| Mind map | 81.93 (-5.75) | Discussions | 84.29 (-3.39) |
| Implications | 85.52 (-2.16) | Teacher-style | 83.68 (-4.00) |
| QA-ct | 84.70 (-2.98) | **Full Set (SPA)** | **87.68** |

### 6.3. Prompt Set Tuning

**Task-specific tuning of the prompt set can further improve performance.** To further analyze the contribution of individual prompts in SPA, we conduct prompt ablation experiments by sequentially removing the lowest-performing prompts identified in Table 4, while keeping the total budget fixed at 22M tokens on SQuAD. As shown in Table 5, tuning the prompt set yields an optimized configuration that achieves 88.19% accuracy on SQuAD, a 0.51% improvement over using all seven prompts.

*Table 5.* **Ablation study over different subsets of prompt strategies** under a fixed budget of 22M tokens on SQuAD. $\mathcal{P}$ denotes the full SPA prompt set, Disc denotes *Discussions*, Key denotes *Key concepts*, Mind denotes *Mind map*, Teach denotes *Teacher-style*, Case denotes *Case Studies*, and Imp denotes *Implications*.

| Prompt Strategies | Acc (SQuAD) |
|---|---|
| $\mathcal{P}$ | 87.68% |
| $\mathcal{P} \backslash \{\text{Key}\}$ | 86.06% |
| $\mathcal{P} \backslash \{\text{Key, Mind}\}$ | 87.27% |
| $\mathcal{P} \backslash \{\text{Key, Mind, Case}\}$ | 87.68% |
| $\mathcal{P} \backslash \{\text{Key, Mind, Case, Teach}\}$ | 85.01% |
| $\mathcal{P} \backslash \{\text{Key, Mind, Case, Teach, Disc}\}$ | **88.19%** |
| $\mathcal{P} \backslash \{\text{Key, Mind, Case, Teach, Disc, QA-ct}\}$ | 85.52% |

**However, the optimal prompt configuration does not transfer across tasks.** We directly transfer the tuned prompt configuration on SQuAD to QuALITY, keeping the total data budget fixed at 27M tokens following Section 5.2. On QuALITY, the full prompt set (SPA) achieves 52.26% accuracy, while the prompt configuration tuned on SQuAD attains 51.51%, resulting in a 0.75% drop. This result suggests that different downstream tasks may favor different forms of knowledge transformation, and that no single prompt configuration is universally optimal.

**SPA therefore serves as a strong default prompt pool for downstream adaptation.** SPA is designed to provide a broadly applicable default prompt set that performs well across tasks. When the downstream task is known, SPA can naturally serve as a prompt pool from which task-specific subsets can be selected, potentially yielding further performance improvements.

## 7. Discussion

**Broader and more challenging scenarios remain to be explored.** We evaluate SPA on three representative knowledge-injection benchmarks: SQuAD (Wikipedia-based QA), QuALITY (long-document comprehension), and MultiHop-RAG (multi-hop reasoning). However, there may exist more challenging scenarios that we do not cover, such as domains requiring intensive numerical reasoning or rapidly evolving knowledge. In addition, whether task-specific tuning of the prompt set can further improve performance across different downstream tasks remains an open direction for future work. Nevertheless, our point is to demonstrate that SPA, as a simple and scalable baseline, is sufficient to outperform prior methods within their original experimental settings.

**SPA is intended as a strong baseline rather than a final solution.** We do not argue that future work should rely exclusively on SPA. Instead, we hope that our results provide the community with insights into the limitations of previous methods and encourage the development of approaches that genuinely improve upon this baseline.

## 8. Conclusion

In this work, we presented SPA, a simple and scalable baseline for knowledge injection. SPA relies on a small, fixed set of carefully designed prompt templates grounded in cognitive learning principles to generate large-scale synthetic corpora for continued pretraining. Across three representative benchmarks, SPA consistently matches or outperforms a range of more complex approaches.

Beyond establishing a baseline, our analyses shed light on key limitations of prior approaches. We empirically demonstrate that RL-based augmentation can suffer from diversity collapse at scale, while multi-stage prompting pipelines may not outperform carefully designed single-stage prompts.

By showing the effectiveness of SPA, we hope that our results will serve as a useful reference point and foundation for future research on scalable, robust, and principled synthetic data generation for continual pretraining. Future work may include how to solve the diversity collapse issues in RL-based approaches and whether we can leverage further insights from cognitive science to design better multi-stage prompting methods for knowledge injection.

## Acknowledgements

We would like to thank Jian Li, Kaiyue Wen, Xinran Gu, and the anonymous reviewers for their insightful comments and feedback that helped improve the quality of this paper. We also thank the broader research community for releasing code, datasets, and evaluation resources that made the comparisons in this work possible.

## Impact Statement

This paper presents work whose goal is to advance scalable knowledge injection for language models. While our method can help models better internalize specialized corpora, large-scale synthetic data generation may also amplify incorrect or hallucinated content if generated data are not carefully checked. In practice, quality control and verification should be applied before using synthetic data for training. Finally, continued pretraining on domain-specific synthetic data may lead to domain overfitting or forgetting of general capabilities, and should be combined with mitigation strategies such as replay. We provide further analysis on SPA's forgetting behavior in Appendix C.5.

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

# A. Prompt templates for Synthetic Data

We list the seven prompt templates used in SPA, ordered according to the learning levels described in Section 3. Each template operates on the same input text, sharing identical grounding constraints while differing only in the learning strategy.

**Prompt Adaptation for Base Models.** For each learning strategy, we provide two prompt variants: one designed for instruction-tuned models and one for base language models. Due to weaker instruction-following ability, base model prompts avoid role-playing instructions, adopt more explicit and imperative formulations, impose stronger constraints to prohibit hallucination, and include an explicit output header to facilitate next-token prediction. Aside from these adaptations, both prompt variants implement the same underlying learning strategy.

## A.1. Key Concepts Prompt

```
Key Concepts (Instruct Model)

System:
You are an assistant tasked with identifying and explaining key concepts
from the provided text. Generate a list of key concepts based on the title
and context provided below. Focus on one concept at a time and explain it
in a clear and detailed way to make it easier to understand and remember.
Each concept explanation should include relevant entities and facts and
preserve important details from the original text.

User:
Title: {title}
Context: {context}
```

```
Key Concepts (Base Model)

Generate a list of key concepts based on the title and text provided below.
Focus on one concept at a time and explain it in a clear and detailed way to
make it easier to understand and remember. Each concept explanation should
include relevant entities and facts and preserve important details from the
original text. Avoid general background knowledge or any unmentioned facts.

Text:
{title}
{context}

Key Concepts and their explanations:
```

## A.2. Mind Map Prompt

```
Mind Map (Instruct Model)

System:
You are an assistant that creates a mind map representation from the
provided text. Create a mind map that organizes the key concepts from the
provided text and represents the relationships between the different concepts.
Explicitly mention relevant entities within the map.

User:
Title: {title}
Context: {context}
```

**Mind Map (Base Model)**

```
Create a mind map that organizes the key concepts from the provided text and
represents the relationships between the different concepts. Explicitly
mention relevant entities within the map. Avoid general background knowledge
or any unmentioned facts. Keep each point concise and avoid repeating any
information.

Text:
{title}
{context}

Mind-map like outline:
```

## A.3. Implications Prompt

**Implications (Instruct Model)**

```
System:
You are an assistant tasked with analyzing the provided passage and
producing a list of implications derived directly or indirectly from
the content.

User:
Title: {title}
Context: {context}
```

**Implications (Base Model)**

```
Read the following text and produce a list of implications derived directly
or indirectly from the content.

Text:
{title}
{context}

Implications:
```

## A.4. QA-ct (QA with Critical Thinking) Prompt

**QA with Critical Thinking (Instruct Model)**

```
System:
You are an assistant tasked with analyzing the provided text and generating
in-depth question-answer pairs based on the provided text. Generate in-depth
question-answer pairs based on the title and text below.
Questions must start with or focus on high-order critical thinking (analysis,
synthesis, evaluation): 'Compare/Contrast', 'Explain the logic of', 'Justify',
'Evaluate the impact of', or 'What if', etc. STRICTLY AVOID: Simple recall,
definitions, or listing facts (e.g., 'What is...', 'List the...').

User:
```

```
Title: {title}
Context: {context}
```

---

**QA with Critical Thinking (Base Model)**

```
Analyze the provided text and generate in-depth question-answer pairs based
on the provided text.
Questions must start with or focus on high-order critical thinking (analysis,
synthesis, evaluation): 'Compare/Contrast', 'Explain the logic of', 'Justify',
'Evaluate the impact of', or 'What if', etc. STRICTLY AVOID: Simple recall,
definitions, or listing facts (e.g., 'What is...', 'List the...'). Avoid
general background knowledge or any unmentioned facts.

Text:
{title}
{context}

Critical thinking question-answer pairs:
```

## A.5. Case Studies prompt

**Case Studies (Instruct Model)**

```
System:
You are an assistant tasked with analyzing the provided text and transforming
it into a structured, professional case study. Generate a formal case-based
writing based on the title and context provided below. Connect the case
facts to the underlying theories or themes in the context. Include the
original title and all key details from the context without changing
their meaning.

User:
Title: {title}
Context: {context}
```

**Case Studies (Base Model)**

```
Analyze the provided text and generate a structured, professional and formal
case study. Include the original title and all key details from the context
without changing their meaning. Avoid general background knowledge or any
unmentioned facts.

Text:
{title}
{context}

Case-based writing from the text:
```

### A.6. Discussions Prompt

---

**Discussions (Instruct Model)**

```
System:
You are an assistant tasked with generating a natural, in-depth discussion
between two readers of a text. Generate a natural, in-depth discussion
between two readers (Person A and Person B) who have both read and are
discussing the provided text. The discussion should remain professional
yet conversational, and stay strictly grounded in the content of
the text. The discussion should explore the core themes, clarify
important ideas, and reflect on implications.

User:
Title: {title}
Context: {context}
```

---

**Discussions (Base Model)**

```
Generate a natural, in-depth discussion between two readers (Person A and
Person B) who have both read and are discussing the provided text. The
discussion should remain professional yet conversational, and stay strictly
grounded in the content of the text. The discussion should explore the core
themes, clarify important ideas, and reflect on implications. Avoid general
background knowledge or any unmentioned facts.

Text:
{title}
{context}

Peer Discussions:
```

---

### A.7. Teacher-style Prompt

---

**Teacher-Style (Instruct Model)**

```
System:
You are an assistant that explains a text as a teacher guiding students to
understand it deeply. Assume the role of a knowledgeable teacher explaining
the article to students who are encountering this text for the first time.
Guide the students step by step through the text and connect different
parts of the text into a coherent understanding. Use clear,
instructional language and explicitly mention relevant entities when
they are introduced.

User:
Title: {title}
Context: {context}
```

---

**Teacher-Style (Base Model)**

```
Explain the following text as a teacher guiding students to understand it
deeply. Assume the role of a knowledgeable teacher explaining the article
```

---

```
to students who are encountering this text for the first time. Guide the
students step by step through the text and connect different parts of the
text into a coherent understanding. Use clear, instructional language and
explicitly mention relevant entities when they are introduced. Avoid general
background knowledge or any unmentioned facts.

Text:
{title}
{context}

Teacher's explanations:
```

## B. Training Details

### B.1. SQuAD

**Hyperparameter tuning.** In the experiment on SQuAD, we observe that training performance is highly sensitive to hyperparameters when the training budget is small. Therefore, for small-scale settings, we perform hyperparameter tuning. To ensure a fair comparison across all token scales, we tuned training hyperparameters separately for each experiment. We summarize our hyperparameter search space in Table 6.

**Hyperparameters for larger-scale experiments.** All other hyperparameters are fixed across experiments: we apply a linear learning rate warmup for 3% of total steps followed by linear decay. We set epochs to 2, context length to 2048 and weight decay to 0. For larger-scale experiments, we fix the peak learning rate to $5 \times 10^{-5}$ and the batch size to 64, keeping others the same as in smaller-scale experiments.

*Table 6.* Hyperparameter search space for experiments on SQuAD. Values in **bold** indicate the hyperparameters used for the largest few experiments within the range of token scales.

| Parameter | Search Space |
|---|---|
| Learning Rate | [4e-5, **5e-5**, 6e-5, 7e-5] |
| Batch Size | [8, **64**] |

### B.2. QuALITY

**Hyperparameters.** In the main paper, all training hyperparameters are fixed across experiments on QuALITY: we apply a linear learning rate warmup for 3% of total steps followed by linear decay. We set epochs to 2, context length to 2048, and weight decay to 0. We fix the peak learning rate to $3 \times 10^{-5}$ and the batch size to 64.

**Training stability analysis.** To further validate the stability of our data's effectiveness, we trained models using SPA's synthetic data on the EntiGraph codebase under the same settings as EntiGraph. For comparison, we also reproduced EntiGraph's results using their official data and training framework, and additionally trained a model on EntiGraph's data using our training framework. All models were trained with 455M synthetic tokens to align with the original settings. The results shown in Table 7 indicate that when using a different training framework, our method's synthetic data still outperforms EntiGraph's data. Moreover, the final result of our model trained on our framework is comparable to that of the model trained on the EntiGraph codebase (57.03% vs 56.64%), demonstrating the robustness of our synthetic data across different training frameworks.

*Table 7.* Training stability comparison on QuALITY across different codebases. Numbers in parentheses indicate results reported in the original EntiGraph paper. All models are trained with 455M synthetic tokens.

| Method | Codebase | QA Accuracy (%) |
|---|---|---|
| EntiGraph | SPA | 55.64 |
| SPA | SPA | **57.03** |
| EntiGraph | EntiGraph | 54.50 (56.22) |
| SPA | EntiGraph | **56.64** |

### B.3. MultiHop-RAG

**Hyperparameters.** In the experiment on MultiHop-RAG, we perform hyperparameter tuning for all methods at the token scale of 15M. We only tune the learning rate, with a search space of [4e-5, 5e-5, 6e-5]. Other parameters are fixed across all methods: We apply a linear learning rate warmup for 3% of total steps followed by linear decay, set epochs to 2, weight decay to 0, context length to 2048, and batch size to 8.

## C. Evaluation Details

### C.1. SQuAD

In our evaluation on SQuAD, we use GPT-4.1 as the LLM judge following SEAL. The grading prompt is shown below.

---
Prompt Template for SQuAD evaluation

```
You are a grading assistant. Your job is to determine whether a student's
answer correctly answers the question based solely on the provided gold
answer. Do not use any outside knowledge. The student answer can include
additional information, but it must at least fully convey the gold answer
and must not contradict it. Ignore style, phrasing, or extra details that
do not affect correctness. Respond ONLY with 'yes' or 'no'.

Question: {question}
Gold answer: {gold}
Student answer: {pred}
Is the student answer correct based solely on the gold answer? Respond
'yes' or 'no'.
```
---

We keep almost the same evaluation setup as SEAL except that we make a modification to better align with our evaluation scenario. Specifically, we consider only the first paragraph of each model prediction during evaluation. We observe that base models often generate irrelevant content after providing their initial answer in the QA-form evaluation. In such cases, LLM-based judges may be misled by the later content and incorrectly mark the prediction as wrong, even when the correct answer appears at the beginning of the model output. We have included some cases below to illustrate this issue.

---
Case: Model output with correct answer followed by irrelevant content

**Question:**
```
Topic: 1973 oil crisis
By which year did Chrysler ended its full sized luxury model?
```

**Model Prediction:**
```
1981

Question: Topic: 1973 oil crisis
What was the result of the 1973 oil crisis?
Answer:
```
---

```
The 1973 oil crisis led to a reduction in the demand for large cars and an
increase in the demand for smaller, more fuel-efficient vehicles
```

**Gold answer:** 1981
**LLM judge** (full output): Wrong
**LLM judge** (first paragraph only, i.e., "1981"): Correct

## C.2. QuALITY

We evaluate on all 4,609 questions associated with the 265 original QuALITY passages used in our experiments. At test time, the original passages are not provided to the adapted model. Following EntiGraph, we use 5-shot chain-of-thought prompting for the multiple-choice QA evaluation.

## C.3. MultiHop-RAG

In our evaluation of MultiHop-RAG, we keep almost the same evaluation setup as the official MultiHop-RAG codebase, except that we make two modifications to better align with our evaluation scenario.

(1) We modify the question prompt to better match our evaluation scenario, which focuses on evaluating base models without providing any original context. The original prompt is designed for instruction-tuned models with access to the context, and is therefore not suitable for our setting. The adapted prompt is shown below.

---

Prompt Template for MultiHop-RAG evaluation

```
Let's answer a question directly and concisely.

Question: {question}

Answer:
```

---

(2) When extracting model predictions, we only consider the first sentence of the model output. We observe that base models often generate long, multi-paragraph responses, where the correct answer may coincidentally appear later in the output even when the initial response fails to answer the question correctly. Including such cases would lead to overly optimistic evaluation results. This issue primarily arises from the weaker instruction-following capabilities of base models compared to instruction-tuned models. We have included some cases below to illustrate this issue.

---

Case 1: Output of model trained on data generated by EntiGraph

**Question:** Who, according to articles in Sporting News, stand to make a profit by predicting outcomes such as a team's lead at the end of a quarter or the total points scored, and can also capitalize on event hype, like putting $130 on the Cowboys to potentially gain $100?

**Model Prediction (excerpt):** The article discusses how the Dallas Cowboys are expected to perform against the Philadelphia Eagles, highlighting the potential for profit in betting scenarios. The mention of the betting public indicates how sportsbooks adjust odds based on the volume of bets placed, which can impact the perceived value of a wager. For instance, if many **bettors** favor the Cowboys, the odds may shift, making it less profitable to bet on them. This dynamic illustrates the relationship between public sentiment and betting strategies, as well as how sportsbooks manage their risk by adjusting odds to balance the bets on both sides...

**Gold answer:** Bettors

---

*Table 8.* **Extended diversity evaluation across datasets.** Synthetic data are generated under the same settings as the main experiments. *Note:* For QuALITY, gpt-oss-120b is used for generation, except for EntiGraph, where we evaluate their released dataset generated by GPT-4o-Turbo. Best results are in **bold**, and second-best are underlined.

| Dataset | Method | Avg CR ↓ | Avg Self-Rep. ↓ | Avg Self-BLEU ↓ | Avg CR: POS ↓ |
|---|---|---|---|---|---|
| SQuAD | Rephrase | 5.1330 | 5.6019 | 0.0014 | 7.0200 |
| | QA | 8.9630 | 6.9882 | 0.0030 | 9.7534 |
| | SEAL | 19.2468 | 8.2446 | 0.0058 | 21.2072 |
| | Active Reading | **4.1788** | **4.5854** | **0.0010** | **5.9388** |
| | SPA | 4.3806 | 4.8207 | **0.0010** | 6.0024 |
| QuALITY | QA | 6.6744 | 6.5268 | 0.0032 | 9.2556 |
| | EntiGraph | 3.9474 | 6.0667 | 0.0016 | 6.4906 |
| | Active Reading | **3.2758** | **3.4497** | **0.0010** | **5.5514** |
| | SPA | 3.7240 | 4.3728 | **0.0010** | 6.0254 |
| MultiHop-RAG | EntiGraph | **4.1760** | 6.2377 | 0.0014 | 6.4530 |
| | Active Reading | 4.5414 | 5.4140 | **0.0010** | 6.6904 |
| | SPA | 4.2580 | **4.4833** | **0.0010** | **6.2620** |

---

**Case 2: Output of model trained on data generated by SPA**

**Question:** Which company, covered by TechCrunch, The Verge, and Fortune, had its largest assets in "unlocked FTT" and "FTT collateral" valued at $5.82 billion, endured a negative balance exceeding the revenue of a related company by $50 million, and took on losses to present a better financial image for that company, while also serving as a customer, payment processor, and market maker within legal boundaries?

**Model Prediction (excerpt):** The company in question is FTX, a cryptocurrency exchange founded by Sam Bankman-Fried. The context indicates that FTX's largest assets were tied to "unlocked FTT," which is a token associated with the FTT cryptocurrency. The valuation of these assets at $5.82 billion highlights the significant financial standing of FTX, but it also reveals a troubling aspect: the company's liabilities exceeded its assets, indicating potential financial instability. The negative balance of $50 million in relation to **Alameda Research**, a trading firm closely linked to FTX, suggests that Alameda was using FTX's resources without adequate compensation, raising questions about financial ethics and transparency...

**Gold answer:** Alameda Research

---

## C.4. Diversity evaluation

The four diversity metrics we use are defined as follows: (1) ***Compression Ratio (CR)*** is computed as the gzip compression ratio of the concatenated corpus, where lower values indicate higher redundancy and thus lower lexical diversity. (2) ***Self-Repetition (Self-Rep.)*** measures the proportion of repeated $n$-grams within the corpus (we use $n = 4$), capturing exact lexical repetition. (3) ***Self-BLEU*** is computed as the average BLEU score of each augmentation against all other augmentations from the same source, quantifying similarity across generated texts. (4) ***Compression Ratio over part-of-speech sequences (CR-POS)*** applies gzip compression to sequences of part-of-speech tags, providing a measure of syntactic redundancy that is independent of lexical content.

We conduct the evaluation on all three benchmarks, as shown in Table 8. For each benchmark, we randomly sample 5 source articles with a fixed sampling seed to ensure reproducibility. To eliminate the effect of different text length on diversity metrics (Salkar et al., 2022; Shaib et al., 2024a), we filter out samples with fewer than 100 words and truncate all remaining augmentations to exactly 100 words. For each source article, we collect 105 rewritings per method and compute diversity scores independently before averaging across articles. The generated rewritings are preprocessed according to each method. For EntiGraph, one summary and 104 relationship rewritings were selected per document. Each relationship rewriting (either pair- or triple-based) consists of entity-level discussions followed by a relationship analysis. Pair- and triple-based rewritings were sampled in equal proportion (52 each), resulting in 105 rewritings per document in total. Other methods treat each output from a given strategy as a single rewriting. For SPA, 15 samples from each of 7 fixed prompts are aggregated per article.

## C.5. General Benchmark Evaluation

To analyze catastrophic forgetting, we conduct additional experiments using lm-eval-harness (Gao et al., 2024) on four general benchmarks: OpenBookQA (Mihaylov et al., 2018), WinoGrande (Sakaguchi et al., 2021), MMLU (Hendrycks et al., 2021), and ARC-C (Clark et al., 2018). We evaluate models after training on two synthetic data scales: SQuAD-based synthetic data with 11M tokens and QuALITY-based synthetic data with 455M tokens. The results are shown in Table 9 and Table 10, respectively.

*Table 9.* **Evaluation on general benchmarks after training on SQuAD-based synthetic data.** SPA shows slight improvements on some benchmarks while largely retaining the performance of the base model. The best average score is in **bold**, and the second best is underlined.

| Method | MMLU | ARC-C | OpenBookQA | WinoGrande | Avg |
|---|---|---|---|---|---|
| Base | 71.93 | 51.37 | 47.20 | 73.32 | 60.96 |
| SEAL | 70.26 | 55.72 | 45.00 | 71.74 | 60.68 |
| Active Reading | 70.27 | 56.31 | 45.60 | 72.53 | **61.18** |
| SPA | 70.51 | 55.12 | 45.60 | 73.16 | 61.10 |

*Table 10.* **Evaluation on general benchmarks after training on QuALITY-based synthetic data.** SPA exhibits the smallest drop in average performance compared to other baselines. The best average score is in **bold**, and the second best is underlined.

| Method | MMLU | ARC-C | OpenBookQA | WinoGrande | Avg |
|---|---|---|---|---|---|
| Base | 62.26 | 54.18 | 44.80 | 73.01 | 58.56 |
| EntiGraph | 55.80 | 48.89 | 43.60 | 68.67 | 54.24 |
| Active Reading | 55.72 | 50.09 | 43.00 | 64.25 | 53.27 |
| SPA | 56.04 | 48.89 | 44.20 | 68.19 | **54.33** |

SPA exhibits less catastrophic forgetting than other baselines. On SQuAD, the model even shows a slight average improvement over the base model. On QuALITY, performance drops after continued pretraining, but the drop under SPA is smaller than for the other baselines, indicating that SPA is no more susceptible to forgetting. We note that forgetting is a known phenomenon in continued pretraining. A standard mitigation strategy is replay, which mixes general pretraining data with domain-specific data. Our setting intentionally excludes replay to allow a clean and direct comparison of synthetic data quality across methods.

# D. Case Study: Examples of Synthetic Data Generated by SPA

We present seven examples of synthetic data generated by applying different SPA learning strategies to rewrite a single text from the SQuAD corpus, demonstrating how the prompt templates reformulate the same content in distinct ways.

> **Original Text (from SQuAD)**
>
> Title: Fresno, California
> Context: The neighborhood of Sunnyside is on Fresnoś far southeast side, bounded by Chestnut Avenue to the West. Its major thoroughfares are Kings Canyon Avenue and Clovis Avenue. Although parts of Sunnyside are within the City of Fresno, much of the neighborhood is a "county island" within Fresno County. Largely developed in the 1950s through the 1970s, it has recently experienced a surge in new home construction. It is also the home of the Sunnyside Country Club, which maintains a golf course designed by William P. Bell.

## D.1. Key Concepts Synthetic Generation

> **Example of *Key concepts* Synthetic Generation**
>
> 1. Fresno, California:
> Fresno is a city located in the Central Valley of California. It is the fifth-largest city in California and the most populous city in the Central Valley region. Fresno is well-known for its agricultural industry, as it is located in the heart of the state's fruit and vegetable production area. The city is also home to several educational institutions, including Fresno State University, and a vibrant arts and cultural scene.
>
> 2. Sunnyside Neighborhood:
> Sunnyside is a neighborhood located in the far southeast part of Fresno. It is bordered on the west by Chestnut Avenue and is bounded by Kings Canyon Avenue and Clovis Avenue to the east and south. Although parts of Sunnyside are within the City of Fresno, much of the neighborhood is a "county island" within Fresno County. Sunnyside is known for its early development, with many of its homes dating back to the 1950s and 1970s.
>
> 3. New Home Construction:
> Sunnyside has recently seen a surge in new home construction. This increase in development is likely due to the fact that many of the homes in the area were built during the 1950s and 1970s and may be in need of modernizing or updating.
>
> 4. Sunnyside Country Club:
> Sunnyside is the home of the Sunnyside Country Club, a private golf club located in the neighborhood. The club's golf course is designed by renowned golf architect William P. Bell. The course is known for its scenic views and challenging layout, making it a popular destination for golf enthusiasts.

## D.2. Mind Map Synthetic Generation

> **Example of *Mind Map* Synthetic Generation**
>
> 1. **Fresno, California**:
> - NEIGHBORHOOD: Sunnyside
> - Boundaries: Chestnut Avenue (West)
> - Major Thoroughfares: Kings Canyon Avenue, Clovis Avenue
> - Territory: Partially within City of Fresno, Majority in Fresno County (county island)
>
> 2. **Sunnyside Neighborhood**:
> - Development Period: 1950s-1970s
> - Recent Development: Surge in new home construction
> - Prominent Features: Sunnyside Country Club
> - Club Details:
> - Golf Course
> - Designed by William P. Bell
>
> This mind-map encapsulates the key points from the provided text, establishing the geographical, historical, and thematic contexts of the Sunnyside neighborhood in relation to Fresno, California.

### D.3. Implications Synthetic Generation

> **Example of *Implications* Synthetic Generation**
>
> 1. Sunnyside is a neighborhood in Fresno, California.
> 2. It is located on the far southeast side of the city.
> 3. It is bounded by Chestnut Avenue to the west.
> 4. Its main thoroughfares are Kings Canyon Avenue and Clovis Avenue.
> 5. Although parts of Sunnyside are within the City of Fresno, much of the neighborhood is in Fresno County.
> 6. The area has experienced significant development between the 1950s and 1970s.
> 7. There has been a recent increase in new home construction within the neighborhood.
> 8. The Sunnyside Country Club is a well-known feature of the area, with a golf course designed by William P. Bell.

### D.4. QA-ct (QA with Critical Thinking) Synthetic Generation

> **Example of *QA-ct* Synthetic Generation**
>
> 1. How has the development of Sunnyside changed over time, and what factors have contributed to its recent surge in new home construction?
> Answer: Sunnyside was largely developed in the 1950s through the 1970s. However, it has recently experienced a surge in new home construction. Factors contributing to this surge may include an increase in demand for housing, population growth, and improvement in the areaś infrastructure.
>
> 2. How does the boundary of Sunnyside relate to its development and its relationship with other areas?
> Answer: Sunnyside is bounded by Chestnut Avenue to the West and has major thoroughfares of Kings Canyon Avenue and Clovis Avenue. This could mean that Sunnyside is somewhat isolated from other parts of Fresno and might have developed in a self-contained manner. This might also have contributed to it becoming a county island within Fresno County.
>
> 3. What does the presence of the Sunnyside Country Club and its golf course designed by William P. Bell suggest about the area?
> Answer: The presence of the Sunnyside Country Club and its golf course designed by William P. Bell suggests that the area is a desirable place for recreation and luxury living, as the golf course is a significant feature that adds to the areaś appeal.
>
> 4. What are the implications of Sunnyside being a "county island" within Fresno County for the residents and local government?
> Answer: The fact that much of Sunnyside is a "county island" within Fresno County means that its governance, services, and regulations may be administered differently than those of Fresno. This could have implications for local residents, who may experience a different level of services or regulations than those in the rest of Fresno. It could also affect the identity of Sunnyside and its relationship to the rest of the city.

### D.5. Case Studies Synthetic Generation

> **Example of *Case Studies* Synthetic Generation**
>
> **Suny-side Neighborhood Case Study**
>
> **Location and Boundaries:**
> The Sunnyside neighborhood is situated in Fresno, California, at its far southeast edge. It is bordered on the west by Chestnut Avenue and traversed by major streets such as Kings Canyon Avenue and Clovis Avenue.

**Geographical and Administrative Status:**
A significant portion of Sunnyside lies within the City of Fresno, while a substantial part resides as a "county island" within Fresno County, signifying its separate administrative status.

**Historical Development:**
Developed primarily during the 1950s and 1970s, Sunnyside experienced a notable boom in residential construction around these decades.

**Recent Developments:**
In recent years, Sunnyside has seen a resurgence in new home construction, contributing to its dynamic development and expanding the neighborhoods living capacity.

*Notable Landmarks and Amenities:**
The Sunnyside Country Club, a prominent landmark within the neighborhood, stands out for its architectural significance as the golf course was designed by renowned architect William P. Bell. This feature underscores the presence of recreational facilities offering community engagement and leisure activities.

**Conclusion:**
This case study elucidates the geographical specifics, historical context, and current developments within the Sunnyside neighborhood, painting a comprehensive picture of its current state and potential future trajectories.

## D.6. Discussions Synthetic Generation

Example of *Discussions* Synthetic Generation

Person A:
Absolutely, the text provides an informative description of Sunnyside in Fresno, California. It outlines the neighborhood's layout and development timeline, marking it as a place that was predominantly developed in the mid-20th century. Additionally, it mentions the notable Sunnyside Country Club with its unique golf course designed by William P. Bell.

Person B:
Yes, I agree with your observations. The mentioning of the Sunnyside Country Club and its golf course designed by William P. Bell is particularly fascinating. It signifies a unique offering in the neighborhood that may not be as readily available in other areas. Also, the fact that Sunnyside is a 'county island' within Fresno County is noteworthy. It indicates that parts of the neighborhood exist independently of the city itself.

Person A:
Yes, the fact that it is a 'county island' seems like an unusual situation. It could be due to historical reasons or early developments in the area that did not get incorporated within the city's growth. Regardless, it adds an interesting layer of complexity to the geography and development of the area.

Person B:
Indeed, this could also be due to planning decisions from the 20th century. In the 1950s through the 70s when Sunnyside was predominantly developed, it's possible that there were specific plans for the area which did not include a direct annexation into Fresno. Also, the recent surge in new home construction suggests an area that has not fully realized its potential for development.

Person A:
You're right. It could very well be due to planning decisions or historical reasons. The surge in new home construction is definitely an exciting development and may well reinvigorate the neighborhood, bringing in fresh energy and possibly new residents.

Person B:
Yes, the new home construction indicates that the neighborhood is seeing positive changes. It implies that Sunnyside is being taken notice of, both by the market and city planners. Hopefully, these developments will lead to improvements in amenities and services in the neighborhood.

Person A:
Definitely, and it's also interesting that a golf course designed by William P. Bell, known for his distinctive golf course designs, is still maintained here. Such a historical asset could be a testament to the neighborhood's heritage and an attraction for visitors and residents alike.

Person B:
Agreed. The Sunnyside Country Club's golf course could be a landmark of sorts, attracting not only golf enthusiasts but also people interested in historic architecture and design. Moreover, it's exciting to see that areas like Sunnyside, which may have been overlooked in the past, are starting to realize their potential for development and growth.

Person A:
Absolutely, it seems that places like Sunnyside are getting the recognition they deserve. The surge in new home construction and the maintenance of historical assets like the golf course designed by William P. Bell all point towards a bright future for this neighborhood.

## D.7. Teacher-style Synthetic Generation

### Example of *Teacher-style* Synthetic Generation

Hello, students. Today we will be exploring the neighborhood of Sunnyside in Fresno, California, and understanding its location, history, and landmarks.

First, let's locate Sunnyside on the map. It is on the far southeast side of Fresno, which is indicated by its proximity to Chestnut Avenue to the west. This geographic information helps us identify where Sunnyside is situated and how it relates to other neighborhoods and landmarks in the area.

Sunnyside has two major thoroughfares that pass through it, Kings Canyon Avenue and Clovis Avenue. Understanding these thoroughfares is essential for knowing the layout of the neighborhood and how different areas are connected.

Another important detail about Sunnyside is that although some parts of the neighborhood are within the City of Fresno, a significant portion is made up of a "county island" within Fresno County. This means that while some areas are governed by the city, others are governed by the county, which may influence local regulations and services.

Now let's talk about the history of Sunnyside. It was largely developed in the 1950s and 1970s. This development history tells us about the age of the neighborhood's homes and buildings, and why the area might have a unique character or architectural style.

In recent years, Sunnyside has experienced a surge in new home construction. This uptick in development might be indicative of its popularity as a desirable place to live or a reflection of changing demographics in the area.

We also need to mention that Sunnyside is home to the Sunnyside Country Club, which has a golf course designed by William P. Bell. This information highlights a notable landmark in the neighborhood and an important recreational feature that residents can enjoy.

In summary, we have learned that Sunnyside is located on the southeast side of Fresno, with major thoroughfares of Kings Canyon Avenue and Clovis Avenue. The neighborhood has experienced significant development in the past and recently saw a surge in new construction. Additionally, Sunnyside is home to the Sunnyside Country Club, which boasts a golf course designed by renowned architect William P. Bell.

This article provides a thorough understanding of the neighborhood of Sunnyside in Fresno, California, including its location, major thoroughfares, development history, recent trends in new construction, and its notable landmark, the Sunnyside Country Club with its golf course designed by William P. Bell.

