# OpenReview forum: "SPA: A Simple but Tough-to-Beat Baseline for Knowledge Injection"
_ICML.cc/2026/Conference — ICML 2026 regular_

### Official Review · Reviewer_1oJJ · 2026-02-23

**Soundness:** 3
**Presentation:** 3
**Significance:** 1
**Originality:** 2
**Overall Recommendation:** 3
**Confidence:** 4

**Summary:**

This paper proposes a simple yet effective baseline (Scaling Prompt-engineered Augmentation, SPA) for injecting domain-specific knowledge into LLMs through synthetic data generation. The authors demonstrate that a straightforward approach using seven human-curated prompt templates can match SOTA performance when scaled, despite recent trends favoring complex RL-based augmentors or multi-stage pipelines. These templates are grounded in cognitive science principles such as concept learning and critical thinking. Through extensive experiments on SQuAD, QuALITY, and MultiHop-RAG benchmarks, the paper demonstrates that SPA maintains strong scaling laws and superior token efficiency in high-data regimes compared to models like SEAL and EntiGraph.

**Compliance With Llm Reviewing Policy:**

Affirmed.

**Final Justification:**

The authors have addressed most of my concerns through the additional scaling experiments and clarifications on the baseline's positioning. However, I still find the methodological contribution somewhat limited as it relies on a fixed set of prompt templates, so I have decided to raise Soundness and Overall Recommendation to 3.

**Key Questions For Authors:**

See the three key Weaknesses above.

**Limitations:**

Yes.

**Strengths And Weaknesses:**

## Strengths

* The work convincingly argues that complexity in augmentation pipelines is not always necessary, demonstrating that a simple, prompt-engineered baseline can outperform RL-based and graph-based approaches. The scaling curves presented illustrate the diminishing returns of RL methods like SEAL at larger token budgets.
* The paper performs token-matched comparisons across multiple benchmarks and model families, which strengthens the credibility of the empirical claims.
* The diversity analysis using standardized metrics provides a plausible explanation for the saturation behavior of RL-based approaches.

## Weaknesses

* SPA relies on a fixed set of manually designed prompt templates, which functions more like a plug-in style prompt engineering enhancement rather than a new learning paradigm. While effective, this design does not demonstrate that SPA can replace RL-based augmentation methods in principle. Instead, the proposed prompt pool could potentially be integrated into RL-based or multi-stage pipelines as a stronger initialization or action space, suggesting complementarity rather than clear methodological superiority.
* The argument that SPA works because of cognitive-strategy-inspired prompts remains largely empirical. However, I think relying on a fixed set of prompts does not fundamentally address the cause of diversity collapse observed in RL-based methods. While SPA shows higher surface-level diversity under current metrics, the template space itself is finite, which may limit the exploration of novel knowledge compositions at larger scales.
* The authors attribute SEAL's performance plateau to a diversity collapse induced by RL optimization. However, SEAL was originally designed to generate only few augmented samples per passage. It is also straightforward to incorporate anti-repetition reward terms into the RL objective, which could potentially mitigate the reported collapse.

---

> ### Author Rebuttal · Authors · 2026-03-31
>
> Dear Reviewer 1oJJ,
>
> Thank you for the encouraging feedback. We appreciate your recognition of **the effectiveness of our simple-yet-strong baseline**, the **token-matched comparisons across benchmarks and models**, and **insightful diversity analysis and scaling curves showing the diminishing returns of RL-based methods**. We address your concerns in detail below.
>
> > ***W1: Novelty***
> >
> - As many prior works are compared against relatively weak baselines such as QA and Rephrase, our main focus is not to propose a novel and complex method but to propose a simple-yet-strong baseline to promote more rigorous evaluation of the actual gains brought by complex methods in the community.
> - Our work also provides insights into the limitations of prior methods(diversity collapse in SEAL and lower strategy effectiveness in Active Reading). Please refer to our response to Reviewer wXs3(W2) for a detailed discussion.
>
> More broadly, we respectfully disagree with the premise that a paper must propose a fundamentally new learning paradigm to make a meaningful contribution. In our view, establishing a strong and simple baseline and providing insights into the limitations of prior methods are valuable contributions to the community in their own right.
>
> > ***W2: Concerns on whether SPA replaces or should be integrated into RL-based methods***
> >
> 1. **SPA is intended as a simple but strong baseline, not a replacement for RL-based methods.** SPA is designed as a baseline rather than a final solution. We do not argue that future work should rely exclusively on SPA. Instead, we hope our results provide the community with insights into the limitations of existing methods and encourage the development of approaches that genuinely improve upon this baseline.
> 2. **The potential integration of SPA into RL-based or multi-stage pipelines is an advantage, not a limitation.** We agree that SPA's design could serve as a stronger initialization or action space within RL-based or multi-stage pipelines. **This is actually an advantage of our method**: SPA can function both as a standalone strong baseline and as a complementary component to enhance other approaches. For example, SPA's prompts could be used to guide strategy selection in Active Reading, potentially yielding further improvements.
>
> > ***W3: Concerns on whether SPA can sustain diversity at larger scales***
> >
> 1. **In our experiments, SPA has already been scaled to very large multiples of the original corpus:** 11M tokens(366x) on SQuAD, 455M tokens(350x) on QuALITY, and 35M tokens(25x) on MultiHop-RAG. In a more real-world scenario where the domain data may contain as many as a few million tokens, augmenting the corpus by 350x already yields over 1 billion tokens.
> 2. **SPA continues to outperform baselines at even larger scales.** To further address your concern, we conduct additional experiments on SQuAD at scales beyond our original largest setting, up to 1000x:
>
>
>     | Scale | SEAL | SPA |
>     | --- | --- | --- |
>     | 11M (366x) | 66.74 | 78.13 |
>     | 22M (733x) | 74.23 | 87.68 |
>     | 30M (1000x) | 75.47 | 89.22 |
>
>     Even at 1000x the original corpus size, SPA continues to outperform SEAL with no sign of saturation, providing strong empirical evidence that its fixed prompt set remains effective at large scales.
>
>
> > ***W4: Concerns on attributing SEAL's plateau to RL-induced diversity collapse, since SEAL was not designed for many augmentations and anti-repetition rewards could mitigate the issue***
> >
> 1. **Evaluating data generation methods with more than 5 augmented samples per passage is a more natural and practical setting.** In SEAL's original design, only 5 augmented samples are generated per passage, which we find insufficient for effective knowledge injection: at this scale the model reaches only 58.2% accuracy. In a more real-world scenario, a model may only become useful once it reaches a sufficiently high accuracy threshold (e.g., 90%). However, it is hard to believe that any method can inject massive domain knowledge with only 5 examples. We therefore argue that a more natural comparison between methods is to evaluate their scaling behavior by generating as many samples as possible and comparing their scaling curves.
> 2. **We agree that anti-repetition rewards are a plausible solution, and our findings provide strong evidence that motivates this direction.** There are many potential directions to explore, such as incorporating anti-repetition reward terms to address diversity collapse, or improving reward design to prevent reward hacking. Our analysis in Sec 6.2 provides empirical evidence that diversity collapse indeed occurs in SEAL at scale, which not only explains its underperformance but also confirms that improving diversity is a well-motivated direction for future work. We hope this finding can guide the community in designing more effective RL objectives for knowledge augmentation.

---

> > ### Author Rebuttal · Reviewer_1oJJ · 2026-04-02
> >
> > The authors have addressed most of my concerns through the additional scaling experiments and clarifications on the baseline's positioning. However, I still find the methodological contribution somewhat limited as it relies on a fixed set of prompt templates, so I have decided to raise Soundness and Overall Recommendation to 3.

---

> > > ### Author Response · Authors · 2026-04-02
> > >
> > > We sincerely appreciate your timely feedback and for raising your score. We would like to address your remaining concern regarding methodological contribution from the following 4 aspects.
> > >
> > > 1. **As many prior works are compared against relatively weak baselines, our main focus is not to propose a novel and complex method but to propose a simple-yet-strong baseline to promote more rigorous evaluation of the actual gains brought by complex methods in the community.** For example, EntiGraph compares against QA and Rephrase, and Active Reading compares against QA. These baselines are relatively easy to surpass, making it difficult to assess the true advantage of the method. We hope our stronger baseline encourages the community to compare methods against stronger baselines, so that the actual gains of methods can be more meaningfully assessed.
> > > 2. **Although SPA relies on a fixed set of prompt templates, it consistently outperforms more complex methods at scale and achieves comparable diversity scores to Active Reading.**  Specifically, with ***only*** 7 human-curated prompts, SPA serves as a very strong baseline. As shown in the table listed in our rebuttal (also pasted below), on SQuAD, the performance does not plateau even at 1000x the original corpus size. Furthermore, SPA has also been validated to be effective on QuALITY (455M tokens, 350x the original corpus) and MultiHop-RAG (35M tokens, 25x the original corpus), which demonstrates its robustness across different settings. SPA also achieves diversity scores comparable to Active Reading(Sec 6.2, also pasted in the second table below), which dynamically generates new strategies as scale increases.
> > >
> > >
> > >     | Scale | SEAL | SPA |
> > >     | --- | --- | --- |
> > >     | 11M (366x) | 66.74 | **78.13** |
> > >     | 30M (1000x) | 75.47 | **89.22** |
> > >
> > >     | Method | CR↓ | NGD↑ | Self-Rep↓ |
> > >     | --- | --- | --- | --- |
> > >     | Active Reading | **4.97** | 1.63 | 6.18 |
> > >     | SPA | 5.02 | **1.68** | **6.05** |
> > > 3. **The simplicity of our method should be viewed as a merit rather than a limitation.** Before introducing additional complexity, it is important to first establish what a simple method can achieve. SPA shows that a carefully designed but straightforward approach can already match or outperform more complex pipelines, suggesting that future complex methods need to demonstrate clear gains over this simple baseline to justify their added complexity. Furthermore, after checking with the AC, we were permitted to share our codebase: https://anonymous.4open.science/r/spacode-0714. ***The complete pipeline makes it easy for the community to reproduce and build upon our baseline.***
> > > 4. **We provide insights into the limitations of prior methods and directions for future work.**
> > >     - **Why RL-based methods underperform.** In Sec 6.2, we evaluate the diversity of synthetic data generated by different methods and find that **SEAL suffers from diversity collapse**: iteratively training on RL-selected data causes the outputs to converge to fixed patterns. This provides a plausible explanation for SEAL's rapidly saturating scaling curve (Figure 2). Future work could explore ways to mitigate diversity collapse in RL-based methods.
> > >     - **Why multi-stage prompting methods underperform.**  To understand why Active Reading underperforms on QuALITY, we conduct an additional per-document experiment comparing the effectiveness of individual prompts/strategies(i.e., training models on each prompt’s synthetic data and evaluating on its corresponding QA set). As shown below, SPA's prompts achieve higher average QA accuracy than Active Reading's prompts across all five documents(45.02% vs. 39.92%), suggesting that **Active Reading yields lower average strategy effectiveness,** thus degrading the overall synthetic corpus quality. Future work could explore combining SPA with Active Reading by using SPA's prompts to guide strategy selection in Active Reading.
> > >
> > >
> > >         | Method\Doc Index | Doc1 | Doc2 | Doc3 | Doc4 | Doc5 | Avg Acc |
> > >         | --- | --- | --- | --- | --- | --- | --- |
> > >         | Active Reading | 23.91 | 47.59 | 38.42 | 54.58 | 35.10 | 39.92 |
> > >         | SPA | **32.14** | **50.42** | **47.37** | **58.57** | **36.61** | **45.02** |
> > >
> > > **Overall, we believe that establishing a simple and strong baseline and providing useful insights into the limitations of prior methods are valuable contributions in their own right.**
> > >
> > > We sincerely appreciate your thoughtful review, which has helped us improve our work and deepen our thinking on future directions for RL-based methods. We hope the above clarifications fully address your remaining concerns, and we would be grateful if you could consider raising your score accordingly. ***Please feel free to follow up and we will be happy to respond.***

---

### Official Review · Reviewer_ygG6 · 2026-03-11

**Soundness:** 3
**Presentation:** 3
**Significance:** 3
**Originality:** 3
**Overall Recommendation:** 4
**Confidence:** 3

**Summary:**

This paper discusses how to efficiently inject knowledge into LLMs in data-scarce domains, enabling the models to internalize small-scale domain-specific corpora rather than overfitting to surface forms. The authors propose SPA (Scaling Prompt-engineered Augmentation) as a simple yet strong baseline method. Its core idea is to design seven prompt templates, combined with large-scale data augmentation. SPA consistently outperforms or matches many more complex prior methods across benchmarks, with particularly significant advantages in high-data regimes.

**Compliance With Llm Reviewing Policy:**

Affirmed.

**Final Justification:**

The rebuttal addressed my concern.

**Key Questions For Authors:**

Among the designed prompt templates, which one has the largest impact on performance for the MultiHop-RAG task? I am particularly interested in this task, and it seems the ablation studies were only conducted on the other two datasets and lack further analysis on the multi-hop setting.

**Limitations:**

yes

**Strengths And Weaknesses:**

**Strengths:**

- The paper is overall really well written and described its method and motivated the problem very well.
- The method is indeed very simple, and the experimental results show surprisingly strong performance.
- The experimental design is rigorous. For example, the authors conduct fair comparisons by strictly controlling the token budget and present clear scaling curves for different methods.

**Weaknesses:**

- The design of the prompt templates feels somewhat heuristic, making it relatively hard to follow.
- The evaluation could be more comprehensive. Currently it is limited to QA tasks and does not cover more challenging scenarios, such as reasoning over very long contexts. Additionally, the experiments on the more difficult MultiHop-RAG task appear relatively limited compared to the other two benchmarks.
- The discussion on generation cost seems insufficient. It remains mostly qualitative, lacking quantitative absolute numbers. (For example, how many tokens/generation steps different methods require to augment the same amount of effective training data using an LLM.)

---

> ### Author Rebuttal · Authors · 2026-03-31
>
> Dear Reviewer ygG6,
>
> Thank you for the encouraging feedback. We appreciate your recognition of the method’s **simple yet surprisingly strong performance**, **clear representation**, and **rigorous and convincing experiments with token-matched comparisons and clear scaling curves**. We address your concerns in detail below.
>
> > ***W1: The design of prompt templates feels somewhat heuristic and hard to follow***
> >
> - **Our prompt templates are grounded in cognitive science and educational psychology.** The prompts in SPA are not arbitrarily chosen. We systematically construct distinct prompt templates guided by principles from cognitive science and educational psychology, with each strategy having a clear counterpart in cited literature. Specifically, we organize the prompts around three levels of learning strategies effective for human learning: *Concept Learning*, *Critical Thinking*, and *Generative Learning*, each of which is defined in detail in Section 3.3.
> - **We provide generation examples to aid understanding.** To help clarify the intent and output of each prompt, we provide generation examples at https://anonymous.4open.science/r/spa-150F/synthetic_data_case.pdf.
>
> > ***W2: Concerns on the limited evaluation scope beyond QA tasks, especially for challenging long-context reasoning scenarios***
> >
> 1. **Our evaluation already includes long-context reasoning.** QuALITY is built on long English passages with an average context length of about 5,000 tokens, consisting mostly of fiction and nonfiction articles.  Keyword search or fact memorization alone is insufficient to answer the questions correctly. Many questions require genuine reasoning: for example, 'why' questions probe characters' internal motivations, requiring the model to infer emotional states from context; symbolism and interpretation questions demand going beyond what is explicitly stated, such as identifying the tone of a story. Thus, QuALITY already includes long-context reasoning challenges.
> 2. **Our focus is on establishing a strong baseline, following the same evaluation protocol as prior methods.** We use benchmarks from prior work: SQuAD (SEAL, PaST), QuALITY (EntiGraph, SoG), and MultiHop-RAG (SoG). While evaluating on more challenging scenarios could be valuable, there are no established baselines for such tasks in the knowledge injection setting, making direct comparison infeasible.
>
> > ***W3: Concerns on insufficient quantitative analysis of generation cost***
> >
>
> We measure generation cost in terms of the number of generator calls required to produce the full token budget for each method on each benchmark (11M tokens on SQuAD, 455M tokens on QuALITY). The results are as follows:
>
> | Benchmark | Method | Generator Calls |
> | --- | --- | --- |
> | SQuAD (11M) | Active Reading | 60,288 |
> | SQuAD | SEAL | 60,032 |
> | SQuAD | SPA | 27,200 |
> | QuALITY (455M) | EntiGraph | 3,990,635 |
> | QuALITY | SPA | 219,950 |
>
> **SPA requires substantially fewer generator calls than other methods.** On SQuAD, Active Reading and SEAL each require approximately 2-3x more generator calls than SPA to produce the same amount of training data. On QuALITY, EntiGraph requires approximately 18x more calls than SPA. Note that generation length naturally varies with the length of the source document, as longer inputs tend to produce longer outputs. It is also worth noting that SEAL incurs additional RL training cost beyond generation (see our response to Reviewer y63q (W7) for details). We will include a more detailed discussion of generation costs in the revised version.
>
> > ***W4: Lack of ablation and prompt-level analysis for MultiHop-RAG***
> >
>
> We conduct an additional ablation study on MultiHop-RAG at the 5M token scale, training Qwen2.5-7B with GPT-4o-mini as the generator, consistent with the experimental settings in the paper. The results are as follows:
>
> | Method | Base | Case Study | Discussion | Implications | Key Concepts | Mind Map | QA-CT | Teacher-Syle | SPA |
> | --- | --- | --- | --- | --- | --- | --- | --- | --- | --- |
> | Acc | 55.00 | 72.67 | 78.31 | 75.12 | 79.66 | 77.21 | 78.43 | 70.59 | 79.66 |
>
> Key Concepts and QA with Critical Thinking (QA-CT) contribute most to performance among individual prompts. Key Concepts prompts the model to identify and explain central concepts in the text, and QA-CT prompts the model to generate question-answer pairs requiring higher-order critical thinking. MultiHop-RAG requires reasoning across multiple passages, where success depends on identifying relevant entities and concepts and reasoning about their relationships. Key Concepts directly supports this by encouraging structured representation of important information, while QA-CT promotes explicit reasoning connections between pieces of knowledge. This may explain why these two prompts are most effective in the multi-hop setting. We will include a more detailed analysis in the revised version.

---

> > ### Author Rebuttal · Reviewer_ygG6 · 2026-04-02
> >
> > Thank you for the thorough rebuttal. I appreciate the clarifications and the effort to run further experiments, and would like to maintain my positive score.

---

> > > ### Author Response · Authors · 2026-04-04
> > >
> > > Dear Reviewer ygG6,
> > >
> > > Thank you for your timely feedback, for recognizing our "thorough rebuttal," and for confirming that all concerns have been fully addressed. We would like to take this opportunity to further elaborate on the principled design of SPA, so that we can offer additional clarity on your concern and share a more complete picture of our work.
> > >
> > > 1. **SPA's prompts are grounded in cognitive science and educational psychology, rather than being heuristic.** Our 7 prompts are not arbitrarily chosen. They are systematically constructed following well-established principles from human learning strategies. They are organized around three levels:
> > >     - ***Concept learning*** is a strategy that requires learners to search for and test attributes that can be used to distinguish exemplars of a concept from non-exemplars. *This level focuses on extracting and organizing the key concepts in the text.*
> > >     - ***Critical thinking*** refers to the process of systematically analyzing facts, evidence, observations, and arguments in order to arrive at well-reasoned conclusions, which has been shown to encourage deep text comprehension. *This level goes beyond surface understanding and requires analyzing and reasoning about the content.*
> > >     - ***Generative learning*** is a strategy that requires learners to actively make sense of learning material so that they can apply it to new situations. *This level further requires constructing new content based on the text, extending the acquired understanding to new scenarios.*
> > >
> > >     Generation examples and are available at https://anonymous.4open.science/r/spa-150F/synthetic_data_case.pdf.
> > >
> > > 2. **Despite its simplicity, SPA is empirically strong and generalizable.** SPA consistently outperforms substantially more complex methods across all three benchmarks at scale. As shown in the table below, on SQuAD, the performance does not plateau even at 1000x the original corpus size. SPA has also been validated on QuALITY (455M tokens, 350x the original corpus) and MultiHop-RAG (35M tokens, 25x the original corpus), which demonstrates its robustness across diverse settings (Sec 5).
> > >
> > >
> > >     | Scale | SEAL | SPA |
> > >     | --- | --- | --- |
> > >     | 11M (366x) | 66.74 | 78.13 |
> > >     | 22M (733x) | 74.23 | 87.68 |
> > >     | 30M (1000x) | 75.47 | 89.22 |
> > >
> > >     Furthermore, SPA achieves comparable diversity scores to Active Reading, while outperforming both the RL-based method SEAL and simple baselines (QA and Rephrase), as shown in the table below.
> > >
> > >     | Method | CR (↓) | NGD (N=4) (↑) | Self-Rep. (↓) |
> > >     | --- | --- | --- | --- |
> > >     | SEAL | 24.42 | 0.20 | 9.45 |
> > >     | Rephrase | 5.77 | 1.40 | 6.75 |
> > >     | QA | 10.07 | 0.67 | 8.21 |
> > >     | Active Reading | **4.97** | 1.63 | 6.18 |
> > >     | SPA | 5.02 | **1.68** | **6.05** |
> > > 3. **A key motivation for SPA is that many prior works are compared against relatively weak baselines, and our main focus is not to propose a novel and complex method but to establish a simple-yet-strong baseline that promotes more rigorous evaluation of the actual gains brought by complex methods.** EntiGraph compares against QA and Rephrase, Active Reading compares against QA, and SEAL compares against training on the original corpus alone. These baselines are relatively easy to surpass, making it difficult to assess the true advantage of these methods. We hope our stronger baseline encourages the community to compare methods against stronger baselines, so that the actual gains of methods can be more meaningfully assessed.
> > > 4. **We provide our entire codebase here to further support the reproducibility  (which has confirmed with AC).** You can access our codebase at: https://anonymous.4open.science/r/spacode-0714. The repository provides the complete pipeline for all three benchmarks, including synthetic data generation, tokenization, training, and evaluation. ***The complete pipeline makes it easy for the community to reproduce and build upon our baseline.***
> > >
> > > Given that all major concerns have been addressed, and considering your earlier recognition that SPA‘s "experimental results show surprisingly strong performance", we would be very grateful if you could consider whether our work merits a higher score (i.e., 5), as this would provide stronger support for our work. ***Please feel free to follow up and we will be happy to respond. Thank you again for your thoughtful review, which has helped us a lot in improving our work.***

---

### Official Review · Reviewer_2dTh · 2026-03-13

**Soundness:** 2
**Presentation:** 2
**Significance:** 3
**Originality:** 3
**Overall Recommendation:** 4
**Confidence:** 2

**Summary:**

LLMs are data-hungry learners, which makes it difficult to finetune them on a small, information-dense dataset. One idea is to synthetically (i.e., through an LLM) expand the dataset to a larger, less dense version, as a form of data augmentation. Several approaches have already been suggested for how to do this concretely: SEAL, EntiGraph, SoG, PaST and Active Reading are the ones considered in this work.

This paper shows that the complexity of these prior approaches has been unwarranted. Indeed, a collection of just 7 handcrafted expansion prompts is shown to be competitive across three commonly used benchmarks and varying expansion token budgets. Ultimately, the paper questions the relevance of existing methods and positions itself as a more suitable baseline for LLM-based data expansion.

**Compliance With Llm Reviewing Policy:**

Affirmed.

**Final Justification:**

I raise my score from 3 to 4 based on the rebuttal.
One concern for the AC: the missing reproducibility details were never reviewed. If those details hide flaws, peer review has no way to catch them.

**Key Questions For Authors:**

My main weaknesses (narrative and reliability of the results) should be addressed mostly through rewriting and adding reproducibility details (incl. the codebase). I have no major substantive questions that need to be answered, I understood the method, the setup and the results.

I do have some smaller questions that I hope the authors can kindly answer for me:
* RAG is discussed in the Related Works. Why is RAG not enough to solve the task? What's the advantage of doing knowledge injection?
* The scaling curves show improved performance for larger token budgets. Is this due to more data or more training steps? Or is the number of training steps kept constant? (this is one of the technical details I'm missing in the paper)
* Why the comparison with PaST and SoG? The code and models are not available. Are their results valid for the exact setup considered in the paper?
* By using a different LLM as generator, the question always remains whether the model is learning from the dataset or from the other LLM (~distillation). What are the 'base' scores of the generators? How does this affect the performance of the various methods? In other words, if I have a really good/bad generator, should I still use SPA or should I look at other methods?
* In SQuAD, why is SEAL worse than the simple baselines of Rephrase and QA? Any insights?
* What are the downsides of using SPA compared to say SEAL or Active Learning? (this could be in the limitations section)

**Limitations:**

The Limitations are not discussed at all. This is really missing from the paper and should definitely be addressed.

**Strengths And Weaknesses:**

__Strengths:__

* The relevance of the application (knowledge injection) is made very clear.
* The premise of the paper is daring: suggesting a very simple method that outperforms existing complex alternatives. Although such papers often risk being received with skepticism, they are vital in keeping the field grounded.
* The writing style is clear and fluent. I could easily read the paper in one go.
* The empirical evaluation is broad, spanning three datasets and several existing methods. Also, the scaling curves nicely demonstrate that SPA's superior performance generalizes across token budgets.

__Weaknesses:__
* The results suggest a consistent victory for SPA, which instinctively raises skepticism as a reader. The primary goal of the paper should therefore be to convince and reassure the reader that all results can be trusted. Right now, the paper spends a lot of space reiterating how good the results are, but this is not helpful. Instead, I'd expect dedicated sections on technical details, reproducibility and proper tuning of baselines. These are all absent, even in the appendix. There is also no limitations section and no attached codebase. If this paper is to replace or supersede existing baselines, people need to be confident that it is indeed consistently better, and that the results were not (accidentally) cherry-picked. Confidence intervals, with results gathered from 5-10 runs, would definitely help, although I realize this is expensive and may not fit the authors' budget.
* The framing of the paper is not entirely consistent. In my summary, I wrote the framing the way I believe it should be, but the paper does not read like this at all. Instead of the "simple beats complex" storyline, it reads more like "yet another prompt engineering paper that scores 2% better", which initially made me dislike the paper. Reasons for this include: prominence of the term "prompt engineering" on the first page and in the method's name, excessive attention to prompt curation (very ad-hoc; based on human learning, which differs strongly from LLM learning), and bold statements to emphasize superior results. I only realized the true intent of the paper through the consistent mention of "SPA is just a simple baseline": this is indeed the main point, but apart from the title and 1-2 paragraphs, this message is not conveyed clearly enough.
* Despite the paper claiming to "provide the community with insights", I'm really missing transferable knowledge beyond the setting-specific observations. Why do these complex methods not work so well? Why does a simple set of prompts beat them? What does this tell us about the task and future methods to solve it? This is all still a mystery to me. Admittedly, such insights are not strictly necessary for the main storyline ("simple beats complex"), but if claimed as contribution, it should be there.
* The Related work section seems relevant enough, but it does not explicitly tie together with the story of the paper. After each paragraph, I'm left wondering "so what?". Also, Section 3.2 "Existing methods" also seems Related Work, but somehow stands separately.
* The ablation study (Table 4) clearly shows that SPA is not the optimal set of prompts for any task. The authors conclude that it is still optimal for QuALITY, but this is without a proper ablation study for that dataset. Ultimately, the question remains: how good is this set of prompts, and could we do better?
* Minor remark: there's some accidental repetition in the writing: "By rearranging the knowledge in this way..." occurs twice in S3.2, and two citations are exactly duplicated (Berglund 2023 & Maini 2024)

__Final verdict:__ Overall, I like the premise of the paper and I would not be against acceptance*. However, I feel that, in its current form, the work will have little impact on the field, as the community will likely dismiss it either as dubious or as a mere prompt engineering paper. A rewriting effort is needed to properly shape the narrative and to provide all necessary implementational details and limitations, in an effort to gain the reader's trust. People will only adopt the method if they believe the results.

I hope the authors will not take these comments personally. Naturally, I do not mean to question the scientific integrity of this work. Trustworthiness is tricky to self-assess as author, because of course you always trust your own results. It just needs to be conveyed more clearly toward the reader.

As a final note, I am no expert in the field and it was hard for me to assess the appropriateness of the selected baselines and the originality of the suggested method. Therefore, I make no critical comments on those.

*: under the assumption that the code will be released upon publication. Otherwise, it's a strong reject for me.

---

> ### Author Rebuttal · Authors · 2026-03-31
>
> Dear Reviewer 2dTh,
>
> Thank you for the encouraging feedback. We appreciate your recognition of our approach’s **simple-yet-strong premise**, and its **broad evaluation with clear scaling curves across benchmarks and methods**. We will revise the paper's positioning and writing accordingly in the revised version. We address your remaining concerns in detail below.
>
> > ***W1: technical details and confidence intervals***
> >
> - **Technical details.** In our experiments, for all baselines, hyperparameters are selected via grid search (learning rate: [3e-5, 4e-5, 5e-5, 6e-5, 7e-5]; batch size: [8, 16, 32, 64]). For the largest-scale experiment in Table 1, we use lr=3e-5 and batch size=64. All experiments use AdamW with linear lr schedule, 3% linear warmup, context length 2048, 2 epochs, and weight decay 0. Full training details will be included in the revised version.
> - **Confidence intervals.** To assess the stability of our results, we conducted additional runs with independently regenerated synthetic data across two data scales:
>
>
>     | Dataset | Mean | Std |
>     | --- | --- | --- |
>     | SQuAD (11M) | 77.62 | ±0.38 |
>     | QuALITY (27M) | 52.25 | ±0.05 |
>
>     These low variance across runs suggests that SPA are robust to both data-generation and training stochasticity. We will include additional confidence intervals for other data scales in the revised version.
>
> - **Codebase:** Due to anonymity constraints, we are checking with the AC to see if we can release our codebase directly. Nevertheless, we provide the code structure overview at https://anonymous.4open.science/r/spa-150F/code_structure.png. The full code will be shared upon acceptance or during discussion if permitted.
>
> > ***W2: framing of the paper's main contribution***
> >
>
> We will revise the paper to more clearly convey: **SPA is a simple baseline outperforming substantially more complex methods.** The reason that we highlight our prompt design process is to clarify that the prompts are grounded in human learning strategies rather than arbitrarily chosen.
>
> > ***W3: Lack of insights into why complex methods underperform***
> >
>
> Please refer to our response to Reviewer wXs3(W2) on why complex methods underperform (diversity collapse in SEAL and lower strategy effectiveness in Active Reading).
>
> > ***W4: Related work is insufficiently connected to main narrative***
> >
>
> We will revise the related work to better connect to the paper's narrative, restructuring it around four topics: 1) small corpus knowledge injection; 2) synthetic data generation, especially the unique challenges of our setting; 3) knowledge editing; and 4) RAG, with a discussion similar to our response to Reviewer y63q(W5). Sec 3.2 is kept separate for providing technical details of evaluated baselines.
>
> > ***W5: insufficient ablation study and concerns on the effectiveness of SPA's prompt set***
> >
>
> Our experiments on QuALITY are not incomplete. We would like to clarify the intent of Table 4. The ablation is conducted exclusively on SQuAD, where we sequentially remove the lowest-performing prompts to find an optimized subset. To evaluate whether this subset generalizes, we transfer it to QuALITY and report its performance.
>
> **SPA is intended as a strong baseline rather than a final solution.** A prompt subset may outperform the full set on specific benchmarks, but SPA remains strong and consistent across settings(Sec 5). We do not claim the full prompt set is always optimal. As noted in Sec 6.1, SPA is designed as a broadly applicable default prompt set, and can serve as a prompt pool when the target task is known. Adding more high-quality prompts to the pool may further improve performance, which we leave for future work.
>
> > ***Q1: advantage of knowledge injection over RAG
> Q3: Validity of comparisons with PaST and SoG
> Q7: limitation analysis***
> >
>
> Please refer to our responses to W5, W2, and W9 of Reviewer y63q for a detailed discussion of Q1, Q3, and Q7, respectively.
>
> > ***Q2: Impact of data quantity vs. training steps***
> >
>
> In our experiments, we fix the total token budget, batch size and context length, and then the number of training steps is set accordingly. So more training data means more steps.
>
> > ***Q4: Role of the generator***
> >
> - **A stronger generator produces higher-quality synthetic data.** We directly evaluate the two generators from Table 2 on QuALITY: GPT-4o achieves 58.01% and GPT-OSS-120B achieves 47.44%, consistent with Table 2 where GPT-4o-generated data yields higher downstream performance.
> - **SPA is robust across generators.** Across different generators, SPA consistently matches or outperforms more complex methods, suggesting robustness to generator quality.
>
> We really learned a lot from your review on how to improve our writing. While we have drafted detailed responses to all your questions, due to space constraints, we were unable to submit all of them. Please feel free to follow up, and we will be happy to respond in a subsequent comment.

---

> > ### Author Rebuttal · Reviewer_2dTh · 2026-04-03
> >
> > Strong rebuttal. The authors addressed all main weaknesses and most questions despite the space constraints.
> >
> > The promised changes (full training details, code release, confidence intervals, revised framing) resolve my two main concerns. With these, a reader could reasonably trust the reported results. I raise my score from 3 to 4.
> >
> > One concern I still have: the reproducibility details were missing during review. If those details contain flaws, peer review has no way to catch them. It means the methodological soundness of the paper is, at this point, not entirely verified.

---

> > > ### Author Response · Authors · 2026-04-04
> > >
> > > Thank you for your timely feedback and for recognizing our rebuttal as a "strong rebuttal". We would like to take this opportunity to further address your remaining concern regarding the reproducibility details.
> > >
> > > 1. **Codebase is now available.** After checking with AC, we were permitted to share our full codebase: https://anonymous.4open.science/r/spacode-0714. The complete pipeline makes it easy for the community to reproduce our results and build upon our baseline.
> > > 2. **Complete experimental details.** To further substantiate the methodological soundness of our work, we provide the full experimental setup below, all of which will be incorporated into the next revision of the paper.
> > >     - ***Data Generation.*** All prompts and generation parameters are available in the codebase. For each benchmark:
> > >         - SQuAD: We use Qwen2.5-7B as the generator and inject 200 contexts from the SQuAD v1 validation set, following exactly the same selection as SEAL.
> > >         - QuALITY: We use GPT-OSS-120B as the generator and inject the full set of 265 documents.
> > >         - Multihop-RAG: We use GPT-4o-mini as the generator and inject the full corpus of 609 documents.
> > >     - ***Training Hyperparameters.*** As stated in our rebuttal: hyperparameters for all baselines are selected via grid search (learning rate: [3e-5, 4e-5, 5e-5, 6e-5, 7e-5]; batch size: [8, 16, 32, 64]). For the largest-scale experiment in Table 1, we use lr=3e-5 and batch size=64. All experiments use AdamW with a linear lr schedule, 3% linear warmup, context length 2048, 2 epochs, and weight decay 0.
> > >     - ***Evaluation Details.***
> > >         - SQuAD: We fully follow SEAL's evaluation setting. We evaluate on 974 questions in a closed-book setting. We use GPT-4.1 as the LLM judge,  and the judge prompt is available at https://anonymous.4open.science/r/spacode-0714/src/utils_tools/utils.py.
> > >         - QuALITY: We follow EntiGraph's evaluation setting, evaluating on 4,609 QA pairs in a closed-book setting, using 5-shot CoT prompting exactly as in EntiGraph. The prompt is available at https://anonymous.4open.science/r/spacode-0714/src/utils_tools/prompt_utils.py.
> > >         - Multihop-RAG: We evaluate on 816 questions, largely following the official evaluation setting. The one deviation is that we modified the question prompt to better match our scenario, which evaluates base models without access to the original context. The original prompt is designed for instruction-tuned models with context provided. The prompt is available at https://anonymous.4open.science/r/spacode-0714/src/eval/eval_mhrag.py.
> > > 3. **Reproducibility was discussed in depth with multiple reviewers during the review stage.** Reviewer y63q asked about the evaluation details for SQuAD (W1&Q1) as well as incomplete training hyperparameters (W4&Q4). Reviewer ygG6 asked about the quantitative details of tokens/generation steps required by different methods (W3). Reviewer 1oJJ asked about the scale of augmentations (W4). All of these reviewers confirmed that our responses addressed the concerns. We will also share the codebase with all other reviewers during the ongoing discussion phase.
> > > 4. **Answers to remaining questions.** Due to space constraints in our rebuttal, we provide our responses to Q5 and Q6 here:
> > >
> > >     > ***Q5:Why SEAL underperforms Rephrase and QA on SQuAD***
> > >     >
> > >
> > >     We evaluate the diversity of data generated by Rephrase and QA using the evaluation pipeline in Sec 6.2. The results show that SEAL still achieves the lowest diversity scores compared to these two simple baselines, which may explain why SEAL underperforms. In contrast, active reading and our method achieved higher diversity scores.
> > >
> > >     | Method | CR↓ | NGD↑ | Self-Rep↓ |
> > >     | --- | --- | --- | --- |
> > >     | SEAL | 24.42 | 0.20 | 9.45 |
> > >     | Rephrase | 5.77 | 1.40 | 6.75 |
> > >     | QA | 10.07 | 0.67 | 8.21 |
> > >     | Active Reading | **4.97** | 1.63 | 6.18 |
> > >     | SPA | 5.02 | **1.68** | **6.05** |
> > >
> > >     > ***Q6:the potential downsides of SPA compared to SEAL or Active Learning***
> > >     >
> > >
> > >     SPA underperforms SEAL under very low token budgets, where RL-based task-specific optimization has a comparative advantage. However, such severely budget-constrained scenarios are rarely practical. In SEAL's design, only 5 augmented samples are generated per passage, which insufficient for effective injection: at this scale the model reaches only 58.2% accuracy. In a more real-world scenario, a model may only become useful once it reaches a sufficiently high accuracy(e.g., 90%). However, it is hard to believe that any method can inject massive domain knowledge with only 5 examples.
> > >
> > >
> > > If you feel that our responses have adequately addressed all your concerns, we would be very grateful if you could consider whether our work merits a higher score (i.e., 5), as this would provide stronger support for our work. Thank you again for your thoughtful and constructive review, which was truly valuable to us.

---

### Official Review · Reviewer_y63q · 2026-03-18

**Soundness:** 3
**Presentation:** 3
**Significance:** 3
**Originality:** 2
**Overall Recommendation:** 4
**Confidence:** 4

**Summary:**

The paper proposes Scaling Prompt-engineered Augmentation (SPA), a simple and scalable baseline for knowledge injection into large language models. SPA uses a fixed set of seven cognitively inspired prompt templates (covering concept learning, critical thinking, and generative learning) to generate large-scale synthetic corpora from small domain-specific documents. Through experiments on SQuAD, QuALITY, and MultiHop-RAG, the authors demonstrate that SPA consistently matches or outperforms more complex augmentation strategies, such as RL-based methods (e.g., SEAL) and multi-stage pipelines (e.g., EntiGraph, Active Reading), under strictly token-matched budgets. Furthermore, the paper identifies that RL-based augmentors experience "diversity collapse" at scale , which provides a mechanism for their diminishing returns, and argues that multi-stage prompting advantages largely disappear with careful prompt engineering.

**Compliance With Llm Reviewing Policy:**

Affirmed.

**Final Justification:**

I would like to thank the authors for their comprehensive and highly constructive rebuttal, as well as for taking the extra step to share the complete anonymous codebase. Your efforts have significantly improved the reproducibility and clarity of the work.

**How the Rebuttal Addressed My Concerns:**
The rebuttal successfully resolved my primary methodological and experimental questions. Specifically:
* The clarification on the SQuAD evaluation protocol (and the codebase release) alleviates my reproducibility concerns.
* The additional experiments on general benchmarks (OpenBookQA, WinoGrande, MMLU, ARC-C) effectively demonstrate that SPA is no more susceptible to catastrophic forgetting than the baselines, and in some cases, performs better.
* The human fact-checking analysis provides a welcome and necessary quantification of hallucination rates, showing them to be relatively infrequent.
* The argument regarding the computational overhead of RL methods (e.g., SEAL requiring 24 GPU hours vs. SPA's 6 hours) is compelling and further strengthens the practical appeal of your approach.

**Final Assessment:**
The paper's strengths are clearly defined: SPA provides a highly practical, logically sound, and empirically validated baseline for knowledge injection. Furthermore, the empirical insight regarding the "diversity collapse" of RL-based methods at scale is a valuable contribution to the community's understanding of data augmentation limits.

However, while the rebuttal was excellent and fully addressed my technical queries, my assessment of the paper's *originality* remains unchanged. The core contribution is primarily empirical, relying heavily on prompt engineering and scaling heuristics rather than introducing fundamentally new algorithmic mechanisms, architectures, or automated hallucination control systems.

Therefore, I am maintaining my score of **Weak Accept (4)**. To be clear, I view this as a positive recommendation.

**Key Questions For Authors:**

1. How exactly is the SQuAD evaluation conducted using the GPT-4.1 judge (e.g., specific prompt, scoring rubric, thresholds)? Can you explain why the no-context parametric model (SPA, 78.13%) substantially surpasses a with-context baseline (68.39%), which is counter-intuitive for reading comprehension tasks?

2. Are the additional training tokens and compute overhead required by RL methods (e.g., policy updates, reward model data) accounted for in the "token-matched" budget comparisons? If not, could you provide compute-normalized comparisons?

3. Do SPA-trained models exhibit catastrophic forgetting on general capabilities? Have you evaluated performance on a small held-out general benchmark (e.g., MMLU subset) before and after continued pretraining?

4. Can you provide the missing training hyperparameters (optimizer, learning rate schedule, batch size, number of updates) across the different data scales?

5. How often do the generated outputs hallucinate or deviate from the source text? Have you conducted any automatic or human fact-checking to quantify this, despite the strict prompt constraints?

**Limitations:**

The authors have not adequately discussed limitations and potential negative societal impact. Their Impact Statement briefly acknowledges that there are societal consequences but actively dismisses the need to discuss them, stating: "There are many potential societal consequences of our work, none of which we feel must be specifically highlighted here.".

**Suggestions for improvement:** The authors should explicitly discuss the risk of large-scale synthetic data reinforcing model hallucinations when prompt constraints fail. They should also address the environmental and computational costs associated with generating hundreds of millions of synthetic tokens, as well as the risk of domain overfitting and catastrophic forgetting of general domain capabilities during the knowledge injection phase.

**Strengths And Weaknesses:**

### Soundness

**Strengths:** The core methodology—prompted synthetic generation followed by continued pretraining—is logically sound and empirically validated. The experimental design is rigorous, featuring token-matched comparisons across multiple strong baselines on three distinct datasets. The ablation studies on prompt combinations (Tables 3 and 4) effectively support the claim that the performance gains stem from the diversity of the prompt set rather than any single template. The diversity metrics (NGD, self-repetition, compression ratio) used to diagnose RL-based diversity collapse corroborate the qualitative narrative well.

**Weaknesses:** The evaluation on SQuAD relies on a GPT-4.1 judge for open-ended answers, but details regarding the judge's prompt, calibration, and acceptance criteria are missing, raising reproducibility concerns. Additionally, the reliance on reported numbers for some baselines (e.g., PaST, SoG) rather than re-running them in the exact same setting limits direct comparability. The paper also lacks an analysis of potential catastrophic forgetting of general capabilities after large-scale synthetic continued pretraining.

### Presentation

**Strengths:** The paper is clearly written, with a well-articulated problem framing and explicit positioning relative to existing RL and multi-stage pipelines. The narrative around scaling and diversity is easy to follow. The prompt templates are thoroughly documented in the appendix, with helpful distinctions between base and instruction-tuned generators.

**Weaknesses:** There are minor typographical inconsistencies across the manuscript (e.g., QuALITY vs. QUALITY, SQuAD vs. SQUAD). Crucial training hyperparameters for the continued pretraining phase (e.g., optimizer, learning rate schedules, batch sizes) are omitted, which hinders full reproducibility.

### Significance

**Strengths:** Establishing a strong, easy-to-implement baseline like SPA is highly valuable for the community, as it provides a practical yardstick to calibrate future claims in knowledge injection. The empirical insight regarding the diversity collapse of RL methods at scale is a significant contribution to the understanding of data augmentation limits.

**Weaknesses:** The lack of head-to-head comparisons with retrieval-augmented generation (RAG) alternatives under a strict inference-time cost budget limits the broader practical significance, despite acknowledging RAG in the related work.

### Originality

**Strengths:** While prompt-based generation is not entirely new, the specific taxonomy of prompts grounded in cognitive science and the demonstration that this simple approach outperforms complex RL/multi-stage pipelines at scale is a novel and useful empirical insight.

**Weaknesses:** The algorithmic novelty is limited; the core contribution relies entirely on prompt engineering and scaling rather than a fundamentally new architecture, loss function, or hallucination control mechanism.

---

> ### Author Rebuttal · Authors · 2026-03-31
>
> Dear reviewer y63q:
>
> Thank you for the encouraging feedback. We appreciate your recognition of our approach’s **logical and empirical soundness**, its **significance as a strong and simple baseline**, and the **novel insights on cognitively grounded prompts and RL/multi-stage pipeline limitations.** We address your concerns in detail below.
>
> > ***W1&Q1: evaluation details for SQuAD***
> >
>
> We follow SEAL’s evaluation exactly. The judge model receives the question, gold answer, and prediction. A prediction is correct if it 1) fully conveys the gold answer 2) without contradiction. Due to anonymity constraints, we are checking with the AC to see if we can release our codebase directly, but we provide the code structure overview at https://anonymous.4open.science/r/spa-150F/code_structure.png. The full code will be shared upon acceptance or during discussion if permitted.
>
> > ***W2: comparability for some baselines***
> >
>
> In Sec 4 of the paper, we note that PaST and SoG do not publicly release their code or data.  However, their reported numbers use the same setting as ours(except that SoG uses a stronger base model yet still underperforms SPA), ensuring a meaningful comparison.
>
> > ***W3&Q3: analysis on potential catastrophic forgetting***
> >
>
> To analyze catastrophic forgetting, we conduct additional experiments on four general benchmarks(OpenBookQA,WinoGrande,MMLU,and ARC-C) under two data scales(SQuAD 11M and QuALITY 455M). Full results for each benchmark are available at https://anonymous.4open.science/r/spa-150F/general_eval_table.pdf
>
> - **SPA exhibits less catastrophic forgetting than other baselines.** On SQuAD, the model even shows *slight average improvement*. On QuALITY, a performance drop occurs. However, the drop under SPA is smaller than in other baselines, indicating SPA is no more susceptible to forgetting.
> - We note that forgetting is a known phenomenon in continual pretraining. A standard mitigation is replay, mixing general pretraining data with domain-specific data. **Our setting intentionally excludes replay to allow a clean, direct comparison of synthetic data quality across methods.**
>
>
>     | Method | SQuAD | Method | QuALITY |
>     | --- | --- | --- | --- |
>     | Base | 60.96 | Base | 58.56 |
>     | SEAL | 60.68 | Entigraph | 54.24 |
>     | Active Reading | **61.18** | Active Reading | 53.27 |
>     | SPA | 61.10 | SPA | **54.33** |
>
> > ***W4&Q4: Incomplete hyperparameters***
> >
>
> Please refer to our response to Reviewer 2dTh(W1) for full hyperparameters.
>
> > ***W5&Q1: Lack of comparison with RAG***
> >
> 1. **Our with-context baseline approximates RAG with oracle retrieval.**  On SQuAD, each document is split into multiple segments. The with-context baseline has access to only one segment at inference time, whereas the SPA model is trained on all segments and therefore develop a more comprehensive understanding of the full document. This may explain why SPA can surpass the with-context baseline.
> 2. **SPA and RAG are complementary rather than competing approaches.** SPA is not intended to replace RAG. Parametric and non-parametric methods serve different purposes and have been shown by Entigraph to have complementary effects.
>
> > ***W6: algorithmic novelty***
> >
>
> Our novelty lies primarily in showing that a simple method grounded in cognitive science can outperform complex methods at scale, and in providing insights into why prior methods underperform. Please refer to our response to Reviewer wXs3(W2) for details.
>
> > ***Q2: compute-normalized comparisons accounting for RL overhead***
> >
>
> In our token-matched experiments, the additional overhead of RL methods is not included. Yet RL-based methods already underperform SPA(Sec 5.1), accounting for this overhead will only widen the gap. To quantify this overhead: SEAL's RL training requires 24 H200 GPU hours, while our largest SQuAD experiments require only 6 GPU hours in total, meaning SEAL's upfront cost alone is several times greater than SPA's entire training cost.
>
> > ***Q5: hallucination frequency and lack of fact-checking of outputs***
> >
>
> We conducted human fact-checking on 50 samples per benchmark, considering both intrinsic(contradicts source) and extrinsic(unverifiable from source) hallucinations.
>
> - **Hallucination frequency varies by generator quality.** For QuALITY(GPT-OSS-120B) and MultiHopRAG(GPT-4o-mini), hallucination rates are low (4%, 2/50), with subtle causal mistakes in single sentences. For SQuAD(Qwen2.5-7B), the rate is higher(12%, 6/50), but most cases involve factually accurate information verifiable from Wikipedia.
> - **Overall, hallucinations are infrequent and mostly benign, especially with stronger generators.**
>
> > ***L1: discussion on potential negative impact and limitations***
> >
>
> We agree that a discussion of potential negative societal impacts is needed and will expand this section in the revised version, covering 1) hallucination risks 2) environmental and computational costs, and 3) catastrophic forgetting.

---

> > ### Author Rebuttal · Reviewer_y63q · 2026-04-02
> >
> > All my questions are addressed. I will maintain my score.

---

> > > ### Author Response · Authors · 2026-04-04
> > >
> > > Dear reviewer y63q,
> > >
> > > Thank you for your timely feedback and for confirming that our rebuttal fully resolved your concerns. Below, we share the full codebase to further support reproducibility, along with a brief summary of the key updates made in response to your feedback.
> > >
> > > **As many of your questions (W1, W4, Q1, and Q4) concern reproducibility, we provide our entire codebase to further support the reproducibility here (which has confirmed with AC).** You can access our codebase at: https://anonymous.4open.science/r/spacode-0714. The repository provides the complete pipeline for all three benchmarks, including synthetic data generation, tokenization, training, and evaluation. In particular, the GPT-4.1 judge prompt used for SQuAD evaluation (***W1&Q1***) can be found at https://anonymous.4open.science/r/spacode-0714/src/utils_tools/utils.py.
> > >
> > > To briefly recap the key improvements made during the rebuttal:
> > >
> > > 1. **Full reproducibility ensured.** We shared the complete codebase (*W1&Q1*) and provided detailed training hyperparameters (*W4&Q4*) .
> > > 2. **Comparisons with all baselines validated (W2, W5&Q1, Q2).**
> > >     - (*W2*) We clarified that PaST and SoG report results under the same setting as ours, ensuring a meaningful comparison.
> > >     - (*W5&Q1*) We also clarified that our with-context baseline approximates RAG with oracle retrieval and that SPA is complementary to RAG rather than competing with it.
> > >     - (*Q2*) We further showed that accounting for RL training overhead only widens the performance gap in favor of SPA, as SEAL's upfront RL cost alone already exceeds SPA's total training cost.
> > > 3. **Additional analysis on catastrophic forgetting** (***W3&Q3***) **and hallucinations** (***Q5***) **:**
> > >     - (*W3&Q3*) We conduct additional experiments on four general benchmarks (OpenBookQA, WinoGrande, MMLU, and ARC-C) under two data scales(SQuAD 11M and QuALITY 455M), showing that SPA exhibits less catastrophic forgetting than other baselines and even improves general capabilities in the SQuAD setting.
> > >     - (*Q5*) Our human fact-checking confirmed that hallucinations in SPA-generated data are infrequent and mostly benign, particularly with stronger generators.
> > >
> > > Given that all major concerns have been addressed, and considering your earlier recognition that SPA is "logically sound and empirically validated" and "provides a highly valuable practical yardstick to calibrate future claims in knowledge injection," we would be very grateful if you could consider whether our work merits a higher score (i.e., 5), as this would provide stronger support for our work. ***Please feel free to follow up and we will be happy to respond. Thank you again for your thoughtful review, which has helped us a lot in improving our work.***

---

### Official Review · Reviewer_wXs3 · 2026-03-23

**Soundness:** 3
**Presentation:** 3
**Significance:** 2
**Originality:** 2
**Overall Recommendation:** 4
**Confidence:** 3

**Summary:**

This is a synthetic data pipeline for generating pretraining data. It uses several different rule-based prompting strategies to produce synthetic data samples. The experiments on multiple QA benchmarks show that SPA achieves better performance than the baselines.

**Compliance With Llm Reviewing Policy:**

Affirmed.

**Final Justification:**

The rebuttal addressed my concern.

**Key Questions For Authors:**

1. What is the difference between this one and prior methods? The paper would benefit from a clearer discussion of how it differs from previous synthetic data generation approaches, such as FLAN and Self-Instruct. In particular, it would be helpful to better articulate the conceptual and practical distinctions beyond the fact that those methods target instruction tuning while this work focuses on pretraining.
2. Can this method be extended to broader domains such as mathematical reasoning or coding? It would be helpful to discuss whether the proposed pipeline remains effective in domains where correctness depends not only on knowledge coverage, but also on reasoning or executable correctness.

**Limitations:**

I feel the novelty is somewhat limited, as a number of prior works have already explored rule-based data generation. And the practical applicability may also be limited, since realistic pretraining scenarios usually require broad-domain knowledge rather than QA-focused data alone.

**Strengths And Weaknesses:**

Strengths:
1. The method is simple and easy to understand. The use of multiple strategies, including concept learning, critical thinking, and case-based analysis, provides a clear framework for generating synthetic samples.

2. The reported results are strong and demonstrate the effectiveness of the approach.

3. The paper is well written and clearly organized.

Weaknesses:
The overall idea is somewhat intuitive and does not appear highly novel, as it is closely related to prior rule-based data generation approaches such as FLAN and Self-Instruct. The main distinction is that earlier methods were designed for instruction tuning, while this work extends a similar philosophy to pretraining.

---

> ### Author Rebuttal · Authors · 2026-03-31
>
> Dear reviewer wXs3:
>
> Thank you for the encouraging feedback. We appreciate your recognition of our method’s **simplicity, yet strong empirical results demonstrating its effectiveness, and the clear presentation.** We address your concerns in detail below.
>
> > ***W1: Insufficient discussion of distinctions from prior rule-based data generation methods.***
> >
> - **FLAN** builds instruction-tuning datasets using task templates applied to *downstream QA pairs*, which aren’t available in the knowledge injection setting when no downstream task is predefined.
> - **Self-Instruct** uses LLM to synthesize instruction-tuning data from seed tasks without grounding in a corpus, making it inapplicable to the knowledge injection setting.
> - **SPA targets knowledge injection by augmenting small domain corpora with diverse synthetic data.**
>
> > ***W2: Clarification on the novelty of SPA***
> >
> 1. **We aim to provide a simple yet strong baseline for knowledge injection.** As many prior methods (e.g., EntiGraph, Active Reading, SEAL) are compared against relatively weak baselines such as QA and Rephrase, our main focus is not to propose a novel and complex method but to propose a simple-yet-strong baseline to promote more rigorous evaluation of the actual gains brought by complex methods in the community.
> 2. **We provide the community with insights into the limitations of prior methods.**
>
>     Beyond establishing a strong baseline, we conduct additional experiments to reveal limitations of prior methods and suggest future directions:
>
>     - **Why RL-based methods underperform.** In Section 6.2, we evaluate the diversity of synthetic data generated by different methods and find that **SEAL suffers from diversity collapse**: iteratively training on RL-selected data causes the outputs to converge to fixed patterns. This provides a plausible explanation for SEAL's rapidly saturating scaling curve (Figure 2). In contrast, SPA's 7 prompts, curated from diverse human learning strategies, avoid this collapse at scales many times larger than the original corpus.
>
>         Future work could explore ways to mitigate diversity collapse in RL-based methods, such as incorporating diversity-promoting RL objectives.
>
>     - **Why multi-stage prompting methods underperform.** To understand why Active Reading underperforms on QuALITY, we conduct an additional per-document experiment comparing the effectiveness of individual prompts/strategies (i.e., training models on each prompt’s synthetic data and evaluating on its corresponding QA set). As shown below, SPA's prompts achieve higher average QA accuracy than Active Reading's prompts across all five documents (45.02% vs. 39.92%), suggesting that **Active Reading yields lower average strategy effectiveness,** thus degrading the overall synthetic corpus quality.
>
>         Future work could explore combining SPA with Active Reading by using SPA's prompts to guide strategy selection in Active Reading. We will provide more details of this experiment in the revised version.
>
>
> | Method\Doc Index | Doc1 | Doc2 | Doc3 | Doc4 | Doc5 | Avg Acc |
> | --- | --- | --- | --- | --- | --- | --- |
> | Active Reading | 23.91 | 47.59 | 38.42 | 54.58 | 35.10 | 39.92 |
> | SPA | **32.14** | **50.42** | **47.37** | **58.57** | **36.61** | **45.02** |
>
> > ***W3: Extension to broader scenarios such as mathematical reasoning or coding.***
> >
> 1. **We focus on establishing a strong baseline, following the same evaluation protocol as prior methods.** We use benchmarks from prior work: SQuAD (SEAL, PaST), QuALITY (EntiGraph, SoG), and MultiHop-RAG (SoG). While evaluating more challenging tasks could be valuable, no established baselines exist for such tasks in the knowledge injection setting, making direct comparison infeasible.
> 2. **Our benchmarks already cover reasoning ability.** For instance, (1) QuALITY is built on long English passages with an average context length of about 5,000 tokens. Keyword search or fact memorization is insufficient to answer the questions correctly. Questions require genuine reasoning: for example, ‘why/reasoning’ questions probe characters’ internal motivations, requiring the model to infer emotional states from context. (2) MultiHop-RAG requires multi-hop reasoning across multiple documents.
>
> > ***W4: Concerns on limited practical applicability and QA-focused data generation.***
> >
> - To clarify, our setting targets **domain-specific knowledge injection** during **continual pretraining**, with **broad applications** in knowledge-intensive fields like medicine, finance, and law.
> - **Our generated data is not solely QA-focused, but diverse in style and structure. Of** SPA’s 7 prompts, only 1 prompt (QA with critical thinking) generates QA-format data; the rest generate discussions, mind-map-style outlines, and other structures. Example generations are available at https://anonymous.4open.science/r/spa-150F/synthetic_data_case.pdf

---

> > ### Author Rebuttal · Reviewer_wXs3 · 2026-04-04
> >
> > Thanks for the reply. I've raised my score to 4.

---

> > > ### Author Response · Authors · 2026-04-07
> > >
> > > Dear Reviewer wXs3,
> > >
> > > Thank you for your encouraging feedback and for confirming that our rebuttal fully addressed your concerns. We would like to take this opportunity to further elaborate on the motivation of SPA and its distinctions from prior rule-based data generation methods, as we believe a clearer understanding of this aspect better reflects the true contribution of our work.
> > >
> > > 1. **SPA is fundamentally different from prior rule-based methods such as FLAN and Self-Instruct.** While these methods share a similar philosophy of using templates to generate synthetic data, they target a fundamentally different setting and are inapplicable to knowledge injection:
> > >     - **FLAN** builds instruction-tuning datasets by applying human-curated task templates to **existing QA pairs**. However, in the knowledge injection setting, no downstream QA pairs are available, so FLAN cannot be easily applied to this setting.
> > >     - **Self-Instruct** uses LLMs to generate new tasks using seed tasks as in-context examples. A related approach in knowledge injection would be to generate new articles from a set of seed articles. However, this approach  may tend to produce articles with entirely new information, rather than reinforcing or refining knowledge within existing articles.
> > >
> > >     In contrast to FLAN and Self-Instruct, **SPA targets the knowledge injection setting**, where the goal is to augment a small domain-specific corpus with diverse synthetic data that reinforces and deepens the model's understanding of the existing content. To this end, SPA employs a principled set of prompts grounded in cognitive science and educational psychology, ensuring that each prompt template is theoretically motivated and collectively covers a diverse range of learning strategies.
> > >
> > > 2. **A key motivation for SPA is that many prior works are compared against relatively weak baselines, and our main focus is not to propose a novel and complex method but to establish a simple-yet-strong baseline that promotes more rigorous evaluation of the actual gains brought by complex methods.** EntiGraph compares against QA and Rephrase, Active Reading compares against QA, and SEAL compares against training on the original corpus alone. These baselines are relatively easy to surpass, making it difficult to assess the true advantage of these methods. We hope our stronger baseline encourages the community to compare methods against stronger baselines, so that the actual gains of methods can be more meaningfully assessed.
> > > 3. **We provide our entire codebase here to further support the reproducibility  (which has confirmed with AC).** You can access our codebase at: https://anonymous.4open.science/r/spacode-0714. The repository provides the complete pipeline for all three benchmarks, including synthetic data generation, tokenization, training, and evaluation. ***The complete pipeline makes it easy for the community to reproduce and build upon our baseline.***
> > >
> > > Given that all major concerns have been addressed, and considering your earlier recognition that SPA's "simplicity, yet strong empirical results demonstrating its effectiveness", we would be very grateful if you could consider whether our work merits a higher score (i.e., 5), as this would provide stronger support for our work. ***Please feel free to follow up and we will be happy to respond. Thank you again for your thoughtful review, which has helped us a lot in improving our work.***

---

### Decision · Program_Chairs · 2026-04-30

**Decision:**

Accept (regular)

**Comment:**

The paper proposes a baseline, called SPA, for injecting domain-specific knowledge into LLMs through synthetic data generation. It is shown that a simple approach, using seven human-curated prompt templates (grounded in cognitive science principles), can match the knowledge injection performance of complex RL-based augmentors or multi-stage pipelines. Experiments on SQuAD, QuALITY, and MultiHop-RAG benchmarks demonstrate that SPA maintains strong scaling laws and can outperform RL-based and graph-based approaches.

The paper makes a strong argument, which could potentially influence both researchers and practitioners in the field, and the reviewers overall found the experiments to be rigorous and the results to be convincing. Also, it seems that most of the reviewers' concerns were resolved during the rebuttal. The only outstanding concern, mentioned by multiple reviewers, is regarding algorithmic novelty. The proposed method practically relies on prompt engineering -- a fixed set of manually crafted prompt templates.

Overall, given the timely argument the paper makes and the convincing results it reports to support it, I believe it could be a valuable addition to the conference.